# Nearly Horizon-Free Offline Reinforcement Learning

**Tongzheng Ren**[1]    **Jialian Li**[2]    **Bo Dai**[3]    **Simon S. Du**[4]    **Sujay Sanghavi**[1, 5]

[1] UT Austin [2] Tsinghua University [3] Google Research, Brain Team
[4] University of Washington [5] Amazon Search

tongzheng@utexas.edu, lijialian7@163.com, bodai@google.com,
ssdu@cs.washington.edu, sanghavi@mail.utexas.edu

## Abstract

We revisit offline reinforcement learning on episodic time-homogeneous Markov Decision Processes (MDP). For tabular MDP with $S$ states and $A$ actions, or linear MDP with anchor points and feature dimension $d$, given the collected $K$ episodes data with minimum visiting probability of (anchor) state-action pairs $d_m$, we obtain nearly horizon $H$-free sample complexity bounds for offline reinforcement learning when the total reward is upper bounded by 1. Specifically:

- For offline policy evaluation, we obtain an $\tilde{O}\left(\sqrt{\frac{1}{Kd_m}}\right)$ error bound for the plug-in estimator, which matches the lower bound up to logarithmic factors and does not have additional dependency on $\mathrm{poly}\,(H, S, A, d)$ in higher-order term.

- For offline policy optimization, we obtain an $\tilde{O}\left(\sqrt{\frac{1}{Kd_m} + \frac{\min(S,d)}{Kd_m}}\right)$ sub-optimality gap for the empirical optimal policy, which approaches the lower bound up to logarithmic factors and a high-order term, improving upon the best known result by [1] that has additional $\mathrm{poly}\,(H, S, d)$ factors in the main term.

To the best of our knowledge, these are the *first* set of nearly horizon-free bounds for episodic time-homogeneous offline tabular MDP and linear MDP with anchor points. Central to our analysis is a simple yet effective recursion based method to bound a "total variance" term in the offline scenarios, which could be of individual interest.

## 1 Introduction

Reinforcement Learning (RL) aims to learn to make sequential decisions to maximize the long-term reward in unknown environments, and has demonstrated success in game-playing [2, 3], robotics [4], and automatic algorithm design [5]. These successes rely on being able to deploy the algorithms in a way that directly interacts with the respective environment, allowing them to improve the policies in a trial-and-error way. However, such direct interactions with real environments can be expensive or even impossible in other real-world applications, *e.g.*, education [6], health and medicine [7, 8], conversational AI [9] and recommendation systems [10]. Instead, we are often given a collection of logged experiences generated by potentially multiple and possibly unknown policies in the past.

This lack of access to real-time interactions with an environment led to the field of *offline reinforcement learning* [11]. Within this, offline policy evaluation (OPE) focuses on evaluating a policy, and offline policy optimization (OPO) focuses on improving policies; both rely only upon the given fixed past experiences without any further interactions. OPE and OPO are in general notoriously difficult, as unbiased estimators of the policy value can suffer from *exponentially* increasing variance in terms of horizon in the worst case [12, 13].

To overcome this "curse of horizon" in OPE, [14, 15] first introduced marginalized importance sampling (MIS) based estimators. They showed that if (1) all of the logged experiences are generated

from the same behavior policy, and (2) the behavior policy is known, then the exponential dependency on horizon can be improved to polynomial dependency. Subsequently, [16–18] showed that the polynomial dependency could be achieved even without assumptions (1) and (2). The basic idea for these MIS-based estimators is estimating the marginal state-action density ratio between the target policy and the empirical data, so as to adjust the distribution mismatch between them. On the algorithmic side, marginal density ratio estimation can be implemented by either plug-in estimators [15, 19], temporal-difference updates [20, 21], or solving a $\min$-$\max$ optimization [16, 17, 22, 18]. These OPE estimators can also be used as one component for OPO, resulting in the algorithms in [23–25], which also inherit the polynomial dependency on horizon.

In a different but related line of work, [26, 27] recently showed that, for the online episodic time-homogeneous tabular Markov Decision Process (MDP) that allows for the interactions with environments, the sample complexity only has $\mathrm{poly}\log$ dependency on the horizon. This motivates us to consider the following question:

*Can offline reinforcement learning escape from the polynomial dependency on the horizon?*

In this paper, we provide an affirmative answer to this question. Specifically, considering the episodic time-homogeneous tabular MDP with $S$ states, $A$ actions and horizon $H$, or the linear MDP with anchor points and feature dimension $d$, assuming the total reward for any episode is upper bounded by $1$ almost surely, we obtain the following nearly $H$-free bounds:

- For offline policy evaluation (OPE), we show that the plug-in estimator has a *finite-sample* error of $\widetilde{O}\left(\sqrt{\frac{1}{Kd_m}}\right)$ (Theorem 1 and 5), where $K$ is the number of episodes and $d_m$ is the minimum visiting probability of (anchor) state-action pairs, that matches the lower bound up to logarithmic factors. We emphasize that the bound has no additional $\mathrm{poly}\,(H, S, A, d)$ dependency in the higher order term, unlike the known results of [19, 24].

- For offline policy optimization (OPO), we show that the policy obtained by model-based planning on empirical MDP has a sub-optimality gap of $\widetilde{O}\left(\sqrt{\frac{1}{Kd_m}} + \frac{\min\{S,d\}}{Kd_m}\right)$ (Theorem 3 and 6), which matches the lower bound up to logarithmic factors and a high-order term. This also improves upon the best known result from [1] by removing additional $\mathrm{poly}\,(H, S, d)$ factors in the main term.

To the best of our knowledge, these are the *first* set of nearly horizon-free bounds for both OPE and OPO on time-homogeneous tabular MDP and linear MDP with anchor points. To achieve such sharp bounds, we propose a novel recursion based method to bound a "total variance" term (introduced below) that is broadly emerged in the offline reinforcement learning, which could be of individual interest.

**Technique Overview.** With a sequence of fairly standard steps in the literature, we can bound the error of the plug-in estimator via terms related to the square root of the "total variance" (also known as the Cramer-Rao type lower bound illustrated in [13]) $\sqrt{\sum_{h\in[H]}\sum_{s,a}\xi_h^\pi(s,a)\mathrm{Var}_{P(s,a)}\left(V_{h+1}^\pi(s')\right)}$ where $\xi^\pi$ is the reaching probability, $P$ is the transition and $V^\pi$ is the value function under policy $\pi$. (For a more formal definition, see Section 3.). An improper bound of this term will introduce unnecessary dependency on the horizon, either in the main term or in the higher-order term. We instead bound this term with a recursive method, by observing that this "total variance" term can be approximately upper bounded by the square root of the "total variance of value square" $\sum_{h\in[H]}\sum_{s,a}\xi_h^\pi(s,a)\mathrm{Var}_{P(s,a)}\left[\left(V_{h+1}^\pi(s')\right)^2\right]$ and some remaining term (see Lemma 1 and Lemma 3 for the detail). Applying this argument recursively, we can finally obtain a $\mathrm{poly}\log H$ upper bound on the total variance, which eventually gets rid of the polynomial dependency on $H$.

The idea of higher order expansion has been investigated in [28, 27]. We notice that the recursion introduced in [28] was designed for the infinite horizon setting, and how to generalize their technique to finite horizon setting is still unclear. Our recursion is conceptually more similar to the recursion in [27]. However, [27] considered the online setting, where we only need to bound the error on the visited state-action pairs. For the offline setting, we need to bound the error on every state-action pair that can be touched by the policy $\pi$. This introduces the reaching probability $\xi^\pi$ in the "total variance" term, which we need to deal with using the MDP structure. As a result, our recursion is significantly different from their counterpart, especially in the case of linear MDP.

**Organization.** Our paper is organized as follows: in Section 2, we review the related literature on offline reinforcement learning, and then briefly introduce the problem we consider in Section 3. In Section 4 and Section 5, we show our results of offline policy evaluation and offline policy optimization on tabular MDP correspondingly, and in Section 6, we show how to generalize our results to linear MDP with anchor points. We finally conclude and discuss our results in Section 7.

## 2 Related Work

In this section, we briefly discuss the related literature in three categories, *i.e.*, offline policy evaluation, offline policy optimization, and horizon-free online reinforcement learning. Notice that, for the setting that assumes an additional generative model, typical model-based algorithms first query equal number of data from each state-action pair, then perform offline policy evaluation/optimization based on the queried data. Thus we view the reinforcement learning with generative model as a special instance of offline reinforcement learning. To make the comparison fair, for method and analysis that do not assume Assumption 1, we scale the error and sample complexity, by assuming per-step reward is upper bounded by $1 - \gamma$ and $1/H$ under infinite-horizon and finite-horizon setting correspondingly.

Table 1: A comparison of existing offline policy evaluation results. The sample complexity in infinite horizon setting is the number of queries of transitions we need while in episodic setting is the number of episodes we need. If Non-Uniform Reward, the MDP we consider satisfies Assumption 1; otherwise, we assume the per-step reward is upper bounded by $1 - \gamma$ and $1/H$ correspondingly.

| Analysis | Setting | Non-Uniform Reward | Sample Complexity |
|----------|---------|--------------------|--------------------|
| [28] | Infinite Horizon Tabular | Yes | $\widetilde{O}\left(\frac{1}{d_m(1-\gamma)\epsilon^2}\right)$ |
| [29] | Infinite Horizon Tabular | Yes | $\Omega\left(\frac{1}{d_m(1-\gamma)\epsilon^2}\right)$ |
| [19] | Finite Horizon time-inhomogeneous Tabular | No | $\widetilde{O}\left(\frac{1}{d_m\epsilon^2} + \frac{\sqrt{SA}}{d_m\epsilon}\right)$ |
| [13] | Finite-Horizon time-inhomogeneous Tabular | No | $\Omega\left(\frac{1}{d_m\epsilon^2}\right)$ |
| This work | Finite Horizon time-homogeneous Tabular/Linear | Yes | $\widetilde{O}\left(\frac{1}{d_m\epsilon^2}\right)$ |
| Lower Bound | Finite Horizon time-homogeneous Tabular/Linear | Yes | $\Omega\left(\frac{1}{d_m\epsilon^2}\right)$ |

**Offline Policy Evaluation.** For OPE in infinite horizon tabular MDP, [28] showed that plug-in estimator can achieve the error of $\widetilde{O}\left(\sqrt{\frac{1}{d_m(1-\gamma)}}\right)$ under Assumption 1, which matches the lower bound in [29] up to logarithmic factors. For OPE in finite horizon time-inhomogeneous tabular MDP, [19, 24] provided an error bound of $\widetilde{O}\left(\sqrt{\frac{1}{Kd_m}} + \frac{\sqrt{SA}}{Kd_m}\right)$ under the uniform reward assumption, which matches the lower bound [13] up to logarithmic factors and an additional higher-order term. We here consider the time-homogeneous MDP, and obtain an error bound of $\tilde{O}(\sqrt{\frac{1}{Kd_m}})$, that does not have the additional $\sqrt{SA}$ in higher-order term, which is different from [19, 24].

Beyond the tabular setting, [30] considered the performance of plug-in estimator with linear function approximation under the assumption of linear MDP without anchor points, and [31, 32] provided more detailed analyses on the statistical properties of different kinds of estimators under different assumptions, which are not directly comparable to our work. Recently, there are also works [e.g. 33] focusing on the interval estimation of the policy for practical application.

**Offline Policy Optimization.** Offline policy optimization for infinite horizon MDP can date back to [35]. [28] recently showed that a perturbed version of model-based planning can find $\epsilon$-optimal policy within $\widetilde{O}\left(\frac{1}{d_m(1-\gamma)\epsilon^2}\right)$ queries of transitions in infinite horizon tabular MDP when the total reward is upper bounded by 1, that matches the lower bound up to logarithmic factors. For the finite horizon time-inhomogeneous tabular MDP setting, [24] showed that model-based planning can identify $\epsilon$-optimal policy with $\widetilde{O}\left(\frac{H}{d_m\epsilon^2}\right)$ episodes, that matches the lower bound for time-inhomogeneous MDP up to logarithmic factors. When it comes to finite horizon time-homogeneous tabular MDP and linrar MDP with anchor points, [1] provided a $\widetilde{O}\left(\frac{\min\{H,S,d\}}{d_m\epsilon^2}\right)$ episode complexity for model-based

Table 2: A comparison of existing offline learning results. The sample complexity in infinite horizon setting is the number of queries of transitions we need while in episodic setting is the number of episodes we need. If Non-Uniform Reward, the MDP we consider satisfies Assumption 1; otherwise, we assume the per-step reward is upper bounded by $1 - \gamma$ and $1/H$ correspondingly.

| Analysis | Setting | Non-Uniform Reward | Sample Complexity |
|---|---|---|---|
| [34] | Infinite Horizon Tabular | No | $\widetilde{O}\left(\frac{1}{d_m(1-\gamma)\epsilon^2}\right)$ |
| [28] | Infinite Horizon Tabular | Yes | $\widetilde{O}\left(\frac{1}{d_m(1-\gamma)\epsilon^2}\right)$ |
| [24] | Finite Horizon time-inhomogeneous Tabular | No | $\widetilde{O}\left(\frac{H}{d_m\epsilon^2}\right)$ |
| [1] | Finite Horizon time-homogeneous Linear | No | $\widetilde{O}\left(\frac{\min\{H,S,d\}}{d_m\epsilon^2}\right)$ |
| [27] | Finite Horizon time-homogeneous Tabular Online | Yes | $\widetilde{O}\left(\frac{SA}{\epsilon^2} + \frac{S^2A}{\epsilon}\right)$ |
| This Work | Finite Horizon time-homogeneous Tabular/Linear | Yes | $\widetilde{O}\left(\frac{1}{d_m\epsilon^2} + \frac{\min\{S,d\}}{d_m\epsilon}\right)$ |
| Lower Bound | Finite Horizon time-homogeneous Tabular/Linear | Yes | $\Omega\left(\frac{1}{d_m\epsilon^2}\right)$ |

planning, which is $\min\{H, S, d\}$ away from the lower bound. We provide a $\widetilde{O}\left(\frac{1}{d_m\epsilon^2} + \frac{\min\{S,d\}}{d_m\epsilon}\right)$ episode complexity, that matches the lower bound up to logarithm factors and a higher order term.

A recent work [36] considered solving the offline policy optimization with model-free $Q$-learning with variance reduction. Although their algorithm can match the sample complexity lower bound $\Omega\left(\frac{H^2}{d_m\epsilon^2}\right)$ when per-step reward is upper bounded by 1, model-free algorithms generally need at least $\Omega(\frac{H^2}{d_m})$ episodes of data to finish the algorithm (a.k.a the sample size barrier in [28]) and it is unclear how to translate their results to the setting with total reward upper bounded by 1.

**Horizon-Free Online Reinforcement Learning.** There are several works obtained nearly horizon-free sample complexity bounds for online reinforcement learning. In the time-homogeneous setting, whether the sample complexity needs to scale polynomially with $H$ was an open problem raised by [37]. The problem was first addressed by [26] who proposed an $\epsilon$-net over the optimal policies and a simulation-based algorithms to obtain a sample complexity that only scales logarithmically with $H$, though their dependency on $S, A$ and $\epsilon$ is not optimal and their algorithm is not computationally efficient. The sample complexity bound was later substantially improved by [27] who obtained an $\widetilde{O}\left(\frac{SA}{\epsilon^2} + \frac{S^2A}{\epsilon}\right)$ bound, which nearly matches the contextual bandits (tabular MDP with $H = 1$) lower bound $\Omega\left(\frac{SA}{\epsilon^2}\right)$ up to an $\frac{S^2A}{\epsilon}$ factor [38]. Their key ideas are (1) a new bonus function and (2) a recursion-based approach to obtain a tight bound on the sample complexity. Such kind of recursion-based approach cannot be directly applied to the offline setting, and we develop a novel recursion method to suit the offline scenarios.

# 3 Problem Setup

**Notation:** Throughout this paper, we use $[N]$ to denote the set $\{1, 2, \cdots, N\}$ for $N \in \mathbb{Z}^+$, $\Delta(E)$ to denote the set of the probability measure over the event set $E$. Moreover, for simplicity, we use $\iota$ to denote $\mathrm{polylog}\,(S, A, H, d, 1/\delta)$ factors (that can be changed in the context), where $\delta$ is the failure probability. We use $\widetilde{O}$ and $\widetilde{\Omega}$ to denote the upper bound and lower bound up to logarithm factors.

## 3.1 Markov Decision Process

Markov Decision Process (MDP) is one of the most standard models studied in the reinforcement learning, usually denoted as $\mathcal{M} = (\mathcal{S}, \mathcal{A}, R, P, \mu)$, where $\mathcal{S}$ is the state space, $\mathcal{A}$ is the action space, $R : \mathcal{S} \times \mathcal{A} \to \Delta(\mathbb{R}^+)$ is the reward, $P : \mathcal{S} \times \mathcal{A} \to \Delta(\mathcal{S})$ is the transition, and $\mu$ is the initial state distribution. We additionally define $r : \mathcal{S} \times \mathcal{A} \to \mathbb{R}^+$ to denote the expected reward.

We focus on the episodic MDP with the horizon[1] $H \in \mathbb{Z}^+$ and time-homogeneous setting[2] that $P$ and $R$ do not depend on the level $h \in [H]$. A (potentially non-stationary) policy $\pi$ is defined as $\pi = (\pi_1, \pi_2, \cdots \pi_H)$, where $\pi_h : \mathcal{S} \to \Delta(\mathcal{A}), \forall h \in [H]$. Following the standard definition, we define the value function $V_h^\pi(s) := \mathbb{E}_{P,\pi}[\sum_{t=h}^H R(s_t, a_t)|s_h = s]$ and the action-value function (i.e. the Q-function) $Q_h^\pi(s, a) := \mathbb{E}_{P,\pi}[\sum_{t=h}^H R(s_t, a_t)|(s_h, a_h) = (s, a)]$, which are the expected cumulative rewards under the transition $P$ and policy $\pi$ starting from $s_h = s$ and $(s_h, a_h) = (s, a)$. Notice that, even though $P$ and $R$ keep invariant under the change of $h$, $V$ and $Q$ always depend on $h$ in the episodic setting, which introduces additional technical difficulties compared with the infinite horizon setting.

The expected cumulative reward under policy $\pi$ is defined as: $v^\pi := \mathbb{E}_\mu [V_1^\pi(s)]$, and our ultimate goal is finding the optimal policy $\pi^*$ of $\mathcal{M}$, which can be written as: $\pi^* = \arg\max_\pi (v^\pi)$. We additionally define the reaching probabilities $\xi_h^\pi(s) = \mathbb{P}_{\mu,P,\pi}(s_h = s), \xi_h^\pi(s, a) = \mathbb{P}_{\mu,P,\pi}(s_h = s, a_h = a)$, that represent the probability of reaching state $s$ and state-action pair $(s, a)$ at time step $h$. Obviously we have $\sum_s \xi_h^\pi(s) = 1, \sum_{s,a} \xi_h^\pi(s, a) = 1, \forall h \in [H]$, and we also have the following relations between $\xi_h^\pi(s)$ and $\xi_h^\pi(s, a)$: $\xi_h^\pi(s, a) = \xi_h^\pi(s)\pi(a|s)$, $\xi_{h+1}^\pi(s') = \sum_{s,a} \xi_h^\pi(s, a)P(s'|s, a)$.

With $\xi_h^\pi$ at hand, we can write $v^\pi$ in an equivalent way: $v^\pi = \sum_{s,a} \left(\left(\sum_{h\in[H]} \xi_h^\pi(s, a)\right) r(s, a)\right)$, which provides a dual perspective on the policy evaluation [18].

## 3.2 Offline Reinforcement Learning

Generally, $P$ and $r$ are not revealed to the learner, which means we can only learn about $\mathcal{M}$ and identify the optimal policy $\pi^*$ with data from different kinds of sources. In offline reinforcement learning, the learner can only have access to a collection of data $\mathcal{D} = \{(s_i, a_i, r_i, s_i')\}_{i\in[n]}$ where $r_i \sim R(s_i, a_i)$ and $s_i' \sim P(\cdot|s_i, a_i)$, that is collected in $K$ episodes with (known or unknown) behavior policy (so that $n = KH$), For simplicity, define $n(s, a)$ as the number of data that $(s_i, a_i) = (s, a)$, while $n(s, a, s')$ is the number of data that $(s_i, a_i, s_i') = (s, a, s')$.

With $\mathcal{D}$, the learner is generally asked to do two kinds of tasks. The first one is the offline policy evaluation (a.k.a off-policy evaluation), that aims to estimate $v^\pi$ for the given $\pi$. The second one is the offline policy optimization, that aims to find the $\hat{\pi}^*$ that can perform well on $\mathcal{M}$. We are interested in the statistical limit due to the limited number of data, and how to approach this statistical limit with simple and computationally efficient algorithms.

# 4 Offline Policy Evaluation

In this section, we first consider the offline policy evaluation for tabular MDP with number of state $S = |\mathcal{S}| < \infty$, number of action $A = |\mathcal{A}| < \infty$, which is the basis for the more general settings. We first introduce the plug-in estimator we consider, which is equivalent to different kinds of estimators that are widely used in practice. Then we describe the assumptions we use, show the error bound of the plug-in estimator, and provide the proof sketch of the error bound.

## 4.1 The Plug-in Estimator

Here we introduce the plug-in estimator. We first build the empirical MDP $\widehat{\mathcal{M}}$ with the data: $\hat{P}(s'|s, a) = \frac{n(s,a,s')}{n(s,a)}, \hat{r}(s, a) = \frac{\sum_{i\in[n]} r_i \mathbf{1}_{(s_i,a_i)=(s,a)}}{n(s,a)}$, where $\mathbf{1}$ is the indicator function. Then we correspondingly define $\hat{Q}_h^\pi, \hat{V}_h^\pi$ and finally the estimator $\hat{v}^\pi, \forall h \in [H]$, by substituting the $P$ and $r$

---

[1] A common belief is that we can always reproduce the results between episodic time-homogeneous setting and infinite horizon setting via substitute the horizon $H$ in episodic setting with the "effective horizon" $\frac{1}{1-\gamma}$ in episodic setting. However, this argument does not always hold, for example the dependency decouple technique used in [34, 28] cannot be directly applied in the episodic setting.

[2] Some previous work consider time-inhomogeneous setting [e.g. 39], where $P$ and $R$ can be varied for different $h \in [H]$. It is noteworthy that we need an additional $H$ factor in the sample complexity to identify $\epsilon$-optimal policy for time-inhomogeneous MDP compared with time-homogeneous MDP. Transforming the improvement analysis from time-homogeneous setting to time-inhomogeneous setting is trivial (we only need to replace $S$ with $HS$), but not vice-versa, as the analysis for time-inhomogeneous setting probably do not exploit the invariant transition sufficiently.

in $Q_h^\pi$, $V_h^\pi$ and $v^\pi$ with $\hat{P}$ and $\hat{r}$. Such computation can be efficiently implemented with dynamic programming. We also introduce the reaching probabilities $\hat{\xi}_h^\pi(s) = \mathbb{P}_{\mu,\hat{P},\pi}(s_h = s), \hat{\xi}_h^\pi(s,a) = \mathbb{P}_{\mu,\hat{P},\pi}(s_h = s, a_h = a)$ in the empirical MDP $\widehat{\mathcal{M}}$, which will be helpful in our analysis.

The plug-in estimator has been studied in [30] under the assumption of linear transition. It's known that the plug-in estimator is equivalent to the MIS estimator proposed in [19] and a certain version of DualDICE estimator with batch update is proposed in [16], due to the observation that $\hat{v}^\pi = \sum_{s,a} \left( \left( \sum_{h \in [H]} \hat{\xi}_h^\pi(s,a) \right) \hat{r}(s,a) \right)$.

## 4.2 Theoretical Guarantee

Here we first summarize the assumptions we use for the tabular MDP.

**Assumption 1** (Bounded Total Reward). $\forall \pi$, *we have* $\sum_{h \in [H]} r_h \leq 1$ *almost surely, where* $s_1 \sim \mu$, $a_h \sim \pi(\cdot|s_h)$, $r_h \sim R(s_h, a_h)$ *and* $s_{h+1} \sim P(\cdot|s_h, a_h)$, $\forall h \in [H]$. *This also means* $\mathbb{P}(r \sim R(s,a)|r > 1) = 0$, $\forall(s,a)$.

This is the key assumption used in [26, 27] to escape the polynomial dependence of horizon in episodic setting. As discussed in [37, 26, 27], this assumption is also more general than the *uniformly bounded reward* assumption: $\forall(s,a), r(s,a) \leq 1/H$. Thus, all of the results in this paper can be generalized to the uniformly bounded reward with a proper scaling of the bounded total reward.

**Assumption 2** (Data Coverage). $\forall(s,a)$, $n(s,a) \geq n d_m$.

This assumption has been used in [19, 24] and is similar to *concentration coefficient* assumption originated from [40]. Intuitively, the performance of the offline reinforcement learning should depend on $d_m$, since the state-action pair with less visitation will introduce more uncertainty.

Notice that, $d_m \in \left(0, (SA)^{-1}\right]$. Assuming access to the generative model, we can query equal number of samples from each state action pair, where $d_m = (SA)^{-1}$. For the offline data sampled with a fixed behavior policy $\pi_{\text{BEH}}$, we can view $d_m \approx \frac{1}{H} \min_{s,a} \sum_{h \in [H]} \xi_h^{\pi_{\text{BEH}}}(s,a)$, which measures the quality of exploration for $\pi_{\text{BEH}}$. When the number of episodes $K = \widetilde{\Omega}\left(1/d_m\right)$, by standard concentration, we know $\min_{s,a} n(s,a) \approx n d_m = HKd_m$.

We remark that, some of the recent Liu et al. [41], Yin et al. [36] define the data coverage via the coverage on the visitation of the optimal policy when considering offline policy optimization. This kind of data coverage can be covered by our Assumption 2, by considering the sub-MDP which only consists of the state-action pair that can be visited by the optimal policy. As we already know that the optimal policy will not leave this sub-MDP, any near-optimal policy on this sub-MDP will be near-optimal on the original MDP, and hence our results can be directly applied under this alternative definition of data coverage.

With these assumptions at hand, we can present our main results.

**Theorem 1.** *Under Assumption 1 and Assumption 2, suppose* $K = \widetilde{\Omega}\left(1/d_m\right)$, *then*
$$|v^\pi - \hat{v}^\pi| \leq \sqrt{\frac{\iota}{Kd_m}}$$
*holds with probability at least* $1 - \delta$, *where* $K$ *is the number of episodes,* $d_m$ *is the minimum visiting probability and* $\iota$ *absorbs the* poly log *factors.*

To demonstrate the tightness of our upper bound, we provide a minimax lower bound in Theorem 2[3].

**Theorem 2.** *There exists a pair of MDPs* $\mathcal{M}_1$ *and* $\mathcal{M}_2$, *and offline data* $\mathcal{D}$ *with* $|\mathcal{D}| = KH$ *and minimum visiting probability* $d_m$, *such that for some absolute constant* $c_0$, *we have*
$$\inf_{\hat{v}^\pi} \sup_{\mathcal{M}_i \in \{\mathcal{M}_1, \mathcal{M}_2\}} \mathbb{P}_{\mathcal{M}_i}\left(|\hat{v}^\pi(\mathcal{D}) - v^\pi| > \frac{c_0}{\sqrt{Kd_m}}\right) > 0.25,$$
*where* $\hat{v}^\pi$ *is any estimator that takes* $\mathcal{D}$ *as input.*

**Remark**   Theorem 1 and 2 show that, even with the simplest plug-in estimator, we can match the minimax lower bound $\Omega\left(\sqrt{\frac{1}{Kd_m}}\right)$ for offline policy evaluation in time-homogeneous MDP up to logarithmic factors. As a result, we can conclude that time-homogeneous MDP is not harder than the bandits in offline policy evaluation.

---

[3]To the best of our knowledge, no lower bound has been provided for finite horizon time-homogeneous setting, so here we provide a minimax lower bound, and the proof can be found in Appendix B.

**Remark** The assumption that $K = \widetilde{\Omega}(1/d_m)$ is a mild and necessary assumption, as with only $o(1/d_m)$ episodes, there can exist some under-explored state-action pair which unavoidably leads to constant error (this is also how we construct the hard instance for the minimax lower bound).

**Remark** Central to our analysis is a recursion based upper bound on the "total variance" term (see Lemma 1), which enables us to have a sharper bound for the plug-in estimator compared with previous work [e.g. 19, 24] that include the unnecessary $\text{poly}(S, A)$ factors in the higher-order term.

### 4.3 Proof Sketch

The detailed proof can be found in Appendix A, and here we sketch our proof in short. With the value difference lemma (see Lemma 4 in the Appendix A), we need to focus on bounding the term:

$$\sum_{h\in[H]} \sum_{s,a} \hat{\xi}_h^\pi(s,a) \left[ \sum_{s'} \left( P(s'|s,a) - \hat{P}(s'|s,a) \right) V_{h+1}^\pi(s') \right],$$

which, by Bernstein's inequality and Cauchy-Schwartz inequality associated with the Assumption 1 and Assumption 2, can be upper bounded by the following term with high probability:

$$\sqrt{\tfrac{\iota}{Kd_m}} \sqrt{\sum_{h\in[H]} \sum_{s,a} \hat{\xi}_h^\pi(s,a) \text{Var}_{P(s,a)}(V_{h+1}^\pi(s'))} + \tfrac{\iota}{Kd_m}$$

To bound the "total variance" term $\sum_{h\in[H]} \sum_{s,a} \hat{\xi}_h^\pi(s,a) \text{Var}_{P(s,a)}[V_{h+1}^\pi(s')]$, we use a novel recursion-based method based on the following observation:

**Lemma 1** (Variance Recursion For Evaluation). *For the value function $V_h^\pi(s)$ induced by any $\pi$, we have that*

$$\sum_{h\in[H]} \sum_{s,a} \hat{\xi}_h^\pi(s,a) \text{Var}_{P(s,a)} \left[ V_{h+1}(s')^{2^i} \right]$$

$$\leq \sum_{h\in[H]} \sum_{s,a} \hat{\xi}_h^\pi(s,a) \left[ \sum_{s'} \left( P(s'|s,a) - \hat{P}(s'|s,a) \right) V_{h+1}(s')^{2^{i+1}} \right] + 2^{i+1} \tag{1}$$

Here the term in (1) can be bounded with $\sum_{h\in[H]} \sum_{s,a} \hat{\xi}_h^\pi(s,a) \text{Var}_{P(s,a)} \left[ V_{h+1}(s')^{2^{i+1}} \right]$, the total variance of higher-order value function via Bernstein's inequality. Thus Lemma 1 can be applied iteratively to obtain a tight bound for the "total variance" term. Define

$$\Delta_1(i) = \left| \sum_{h\in[H]} \sum_{s,a} \hat{\xi}_h^\pi(s,a) \left[ \sum_{s'} \left( P(s'|s,a) - \hat{P}(s'|s,a) \right) V_{h+1}^\pi(s')^{2^i} \right] \right|.$$

Applying Lemma 1 with $V_h^\pi(s)$, we can write the following recursion:

$$\Delta_1(i) \leq \sqrt{\tfrac{\iota}{Kd_m} \left( \Delta_1(i+1) + 2^{i+1} \right)} + \tfrac{\iota}{Kd_m}.$$

Solve this recursion, and we can obtain the results in Theorem 1.

## 5 Offline Policy Optimization

In this section, we further consider the offline policy optimization for tabular MDP, which is the ultimate goal for offline reinforcement learning. We first introduce the model-based planning algorithm, which is probably the simplest algorithm for offline policy optimization. Then we analyze the performance gap between the policy obtained by model-based planning and the optimal policy.

### 5.1 Model-Based Planning

We consider the optimal policy on the empirical MDP $\widehat{\mathcal{M}}$, which can be defined as

$$\hat{\pi}^* = \arg\max_\pi \hat{v}^\pi. \tag{2}$$

Here $\hat{\pi}^*$ can be obtained by dynamic programming with the empirical MDP, which is also known as model-based planning. We remark that our analysis is independent to the algorithm used for solving (2). In other words, the result also applies to the optimization-based planning with the empirical MDPs [42, 43], as long as it solves (2).

## 5.2 Theoretical Guarantee

Theorem 3 provides an upper bound on the sub-optimality of $\hat{\pi}^*$.

**Theorem 3.** *Under Assumption 1 and Assumption 2, suppose that $K = \widetilde{\Omega}\left(1/d_m\right)$, and then*

$$\left|v^{\pi^*} - v^{\hat{\pi}^*}\right| \leq \sqrt{\frac{\iota}{Kd_m}} + \frac{S\iota}{Kd_m},$$

*holds with probability at least $1 - \delta$, where $K$ is the number of episodes, $d_m$ is the minimum visiting probability and $\iota$ absorbs the* poly log *factors.*

We also provide a minimax lower bound under the finite horizon time-homogeneous setting.

**Theorem 4.** *There exists a pair of MDPs $\mathcal{M}_1$ and $\mathcal{M}_2$, and offline data $\mathcal{D}$ with $|\mathcal{D}| = KH$ and minimum state-action pair visiting frequency $d_m$, such that for some absolute constant $c_0$, we have*

$$\inf_{\hat{v}^\pi} \sup_{\mathcal{M}_i \in \{\mathcal{M}_1, \mathcal{M}_2\}} \mathbb{P}_{\mathcal{M}_i}\left(\left|v^{\hat{\pi}(\mathcal{D})} - v^{\pi^*}\right| > \frac{c_0}{\sqrt{Kd_m}}\right) > 0.25,$$

*where $\hat{\pi}$ is any planning algorithm that takes $\mathcal{D}$ as input.*

**Remark** Theorem 3 provides a bound approaching the minimax lower bound in Theorem 4 up to logarithmic factors and a higher-order term, which shows that the error of offline policy optimization does not scale polynomially on the horizon. Notice that if $d_m = \Omega\left(\frac{1}{SA}\right)$, we can obtain an error bound of $\widetilde{O}\left(\sqrt{\frac{SA}{K}} + \frac{S^2A}{K}\right)$, which can be translated to sample complexity $\widetilde{O}\left(\frac{SA}{\epsilon^2} + \frac{S^2A}{\epsilon}\right)$ that matches the best known result of sample complexity for online finite-horizon time-homogeneous setting [27]. We conjecture that the additional $S$ factor in the higher-order term is only an artifact (see Lemma 2) and can be eliminated with more delicate analysis. We leave this as an open problem.

**Remark** There are also works considering local policy optimization [e.g. 44–46] when the offline data are not sufficient exploratory. We want to emphasize that, as Theorem 4 suggests, to identify the global optimal policy, we need the offline data sufficient exploratory.

## 5.3 Proof Sketch

The detailed proof can be found in Appendix A, and here we sketch our proof in short. Notice that

$$v^{\pi^*} - v^{\hat{\pi}^*} = v^{\pi^*} - \hat{v}^{\pi^*} + \underbrace{\hat{v}^{\pi^*} - \hat{v}^{\hat{\pi}^*}}_{\leq 0} + \hat{v}^{\hat{\pi}^*} - v^{\hat{\pi}^*}$$

$$\leq \underbrace{v^{\pi^*} - \hat{v}^{\pi^*}}_{\text{Error on Fixed Policy}} + \underbrace{\hat{v}^{\hat{\pi}^*} - v^{\hat{\pi}^*}}_{\text{Error on Data−Dependent Policy}}. \tag{3}$$

We can directly apply Theorem 1 to bound the error on fixed policy. For the error on data-dependent policy, since the policy $\hat{\pi}^*$ depends on data, we need to consider

$$\sum_{h \in [H]} \sum_{s,a} \hat{\xi}_h^\pi(s,a)\left[\sum_{s'}\left(P(s'|s,a) - \hat{P}(s'|s,a)\right)V_{h+1}^{\hat{\pi}^*}(s')\right]$$

$$= \sum_{h \in [H]} \sum_{s,a} \hat{\xi}_h^\pi(s,a)\left[\sum_{s'}\left(P(s'|s,a) - \hat{P}(s'|s,a)\right)V_{h+1}^{\pi^*}(s')\right] \tag{4}$$

$$+ \sum_{h \in [H]} \sum_{s,a} \hat{\xi}_h^\pi(s,a)\left[\sum_{s'}\left(P(s'|s,a) - \hat{P}(s'|s,a)\right)\left(V_{h+1}^{\hat{\pi}^*}(s') - V_{h+1}^{\hat{\pi}^*}(s')\right)\right]. \tag{5}$$

As $\pi^*$ is independent of $\hat{P}$, the term in (4) can be similarly handled with the techniques used in offline policy evaluation. However, due to the dependency of $\hat{\pi}^*$ and $\hat{P}$, we need to deal with the term in (5) more carefully. To decouple the dependency of $\hat{\pi}^*$ and $\hat{P}$, we first introduce the following lemma:

**Lemma 2.** *$\forall V_h(s) \in [0,1]$, $\forall h \in [H], s \in \mathcal{S}$, then we have that with high probability,*

$$\left|\sum_{s'}\left(P(s'|s,a) - \hat{P}(s'|s,a)\right)V_h(s')\right| \leq \sqrt{\frac{S \cdot \mathrm{Var}_{P(s,a)}[V_h(s')]\iota}{n(s,a)}} + \frac{S\iota}{n(s,a)}.$$

**Remark** Lemma 2 has been widely used in the design and analysis of online reinforcement learning algorithms, [e.g. 27]. It holds even $V_h(s)$ depends on $\hat{P}(s'|s,a)$, however, at the cost of an additional $S$ factor. This is the source of the additional $S$ factor in the higher-order term, and we believe a more fine-grained analysis can help avoid this additional $S$ factor.

We also have the following recursion for (4) and (5):

**Lemma 3** (Variance Recursion For Optimization). *For $V_h(s) = V_h^{\pi^*}(s)$ and $V_h(s) = V_h^{\pi^*}(s) - V_h^{\hat{\pi}^*}(s)$, we have that*

$$\sum_{h \in [H]} \sum_{s,a} \hat{\xi}_h^{\hat{\pi}^*}(s,a) \mathrm{Var}_{P(s,a)} \left[ V_{h+1}(s')^{2^i} \right]$$

$$\leq \sum_{h \in [H]} \sum_{s,a} \hat{\xi}_h^{\hat{\pi}^*}(s,a) \left[ \sum_{s'} \left( P(s'|s,a) - \hat{P}(s'|s,a) \right) V_{h+1}(s')^{2^{i+1}} \right] \tag{6}$$

$$+ 2^{i+1} \left[ \sum_s \mu(s) V_1(s) + \sum_{h \in [H]} \sum_{s,a} \hat{\xi}_h^{\hat{\pi}^*}(s,a) \sum_{s'} \left[ P(s'|s,a) - \hat{P}(s'|s,a) \right] V_{h+1}(s') \right]. \tag{7}$$

Denote

$$\Delta_2(i) = \left| \sum_{h \in [H]} \sum_{s,a} \hat{\xi}_h^{\pi}(s,a) \left[ \sum_{s'} \left( P(s'|s,a) - \hat{P}(s'|s,a) \right) V_{h+1}^{\pi^*}(s')^{2^i} \right] \right|,$$

$$\Delta_3(i) = \left| \sum_{h \in [H]} \sum_{s,a} \hat{\xi}_h^{\pi}(s,a) \left[ \sum_{s'} \left( P(s'|s,a) - \hat{P}(s'|s,a) \right) \left( V_{h+1}^{\pi^*}(s') - V_{h+1}^{\hat{\pi}^*}(s') \right)^{2^i} \right] \right|.$$

Applying Bernstein inequality or Lemma 2, and then Lemma 3, we have that:

$$\Delta_2(i) \leq \sqrt{\frac{\iota}{Kd_m} \left( \Delta_2(i+1) + 2^{i+1}(v^{\pi^*} + \Delta_2(0)) \right)} + \frac{\iota}{Kd_m}$$

$$\Delta_3(i) \leq \sqrt{\frac{S\iota}{Kd_m} \left( \Delta_2(i+1) + 2^{i+1}(v^{\pi^*} - v^{\hat{\pi}^*} + \Delta_3(0)) \right)} + \frac{S\iota}{Kd_m}.$$

With Assumption 1, we know $v^{\pi^*} \leq 1$. Also, with (3) and (4), we have that $v^{\pi^*} - v^{\hat{\pi}^*} \leq \Delta_3(0) + O\left( \sqrt{\frac{\iota}{Kd_m}} \right)$. Solve this recursion, then we can obtain the results in Theorem 3.

## 6 Extensions to Linear MDP with Anchor Points

In this section, we first introduce the definition of the linear MDP with anchor points [47, 1], and then generalize our results of offline policy evaluation and optimization to this setting.

**Definition 1** (Linear MDP with Anchor Points [47, 1]). *For the MDP $\mathcal{M} = (\mathcal{S}, \mathcal{A}, R, P, \mu)$, assume there is a feature map $\phi : \mathcal{S} \times \mathcal{A} \to \mathbb{R}^d$, such that $r$ and $P$ admits a linear representation:*

$$r(s,a) = \langle \phi(s,a), \theta_r \rangle \quad P(s'|s,a) = \langle \phi(s,a), \mu(s') \rangle,$$

*where $\mu$ is an unknown (signed) measure of $\mathcal{S}$. Furthermore, we assume there exists a set of anchor state-action pairs $\mathcal{K}$, such that $\forall (s,a) \in \mathcal{S} \times \mathcal{A}$, $\exists \{ \lambda_k^{s,a} \}_{k \in \mathcal{K}}$,*

$$\phi(s,a) = \sum_{k \in \mathcal{K}} \lambda_k^{s,a} \phi(s_k, a_k), \quad \sum_{k \in \mathcal{K}} \lambda_k^{s,a} = 1, \quad \lambda_k^{s,a} \geq 0, \forall k \in \mathcal{K}.$$

With the definition of Linear MDP as well as the anchor points assumption, we can find that $r(s,a) = \sum_{k \in \mathcal{K}} \lambda_k^{s,a} r(s_k, a_k)$, $P(s'|s,a) = \sum_{k \in \mathcal{K}} \lambda_k^{s,a} P(s'|s_k, a_k)$, which can lead to an empirical estimation of $\hat{P}$ and $\hat{r}$ by replacing $\{ r(s_k, a_k) \}_{k \in \mathcal{K}}$, $\{ P(s'|s_k, a_k) \}_{k \in \mathcal{K}}$ with the empirical counterpart $\{ \hat{r}(s_k, a_k) \}_{k \in \mathcal{K}}$, $\{ \hat{P}(s'|s_k, a_k) \}_{k \in \mathcal{K}}$ estimated from the offline data. Following [47, 1], we additionally make the following assumption on the offline data:

**Assumption 3** (Anchor Point Data [47, 1]). *Assume* $|\mathcal{K}| = d$. *For the offline data* $\mathcal{D} = \{(s_i, a_i, r_i, s'_i)\}_{i \in [n]}$, $(s_i, a_i) \in \{(s_k, a_k)\}_{k \in \mathcal{K}}$, $\forall i \in [n]$. *Furthermore,* $\forall k \in \mathcal{K}$, $n(s_k, a_k) \geq nd_m$.

We now present the main theorem on the offline policy evaluation and offline policy improvement on the linear MDP with anchor points.

**Theorem 5.** *Under Assumption 1 and Assumption 3, suppose* $n = \widetilde{\Omega}(H/d_m)$, *and then for a given policy* $\pi$, *the plug-in estimator* $\hat{v}^\pi$ *satisfies*

$$|v^\pi - \hat{v}^\pi| \leq \sqrt{\frac{H\iota}{nd_m}}$$

*with probability at least* $1 - \delta$, *where* $n$ *is the number of offline data,* $d_m$ *is the minimum visiting probability of anchor points and* $\iota$ *absorbs the* polylog *factors.*

**Theorem 6.** *Under Assumption 1 and Assumption 3, suppose that* $n = \widetilde{\Omega}(H/d_m)$, *and then for the* $\hat{\pi}^*$ *obtained by model-based planning,*

$$\left|v^{\pi^*} - v^{\hat{\pi}^*}\right| \leq \sqrt{\frac{H\iota}{nd_m}} + \frac{dH\iota}{nd_m},$$

*holds with probability at least* $1 - \delta$, *where* $n$ *is the number of offline data,* $d$ *is the feature dimension,* $d_m$ *is the minimum visiting probability of anchor points and* $\iota$ *absorbs the* polylog *factors.*

Proof of both theorems can be found in Appendix C. From a high-level perspective, we observe that for the unseen state-action pair, we still have Bernstein-type concentration bound and a lemma similar to Lemma 2. Hence we can apply our recursion introduced in Lemma 1 and use the similar techniques for tabular MDP to obtain the desired results. We want to remark that such results demonstrate the broad applicability of our recursion-based analysis in different kinds of offline scenarios.

**Remark** Compared with the results in [1], we remove the additional dependency of $\min\{H, |\mathcal{S}|, d\}$ in the main term and approach the optimal complexity shown in the [47] up to logarithmic factors, which shows that model-based planning is minimax optimal for Linear MDP with anchor points and demonstrates the effectiveness of our recursion-based analysis.

**Remark** Here we do not directly replace the term $n/H$ with $K$, as we make relatively strong assumption on all of the offline data are sampled from the anchor points. One potential question is how general the anchor point assumption is. We notice that, as [30] showed, the popular FQI algorithm for linear MDP provides an estimate of $\hat{P}$ as

$$\hat{P}(s'|s, a) = \phi(s, a)^\top \hat{\Lambda}^{-1} \sum_{i \in [n]} \phi(s_i, a_i) \mathbf{1}_{s'=s'_i},$$

where $\hat{\Lambda} = \sum_{i \in [n]} \phi(s_i, a_i) \phi(s_i, a_i)^\top$. As $\sum_{s'} P(s'|s, a) = 1$, by Definition 1,

$$1 = \sum_{s'} P(s'|s, a) = \phi(s, a)^\top \sum_{s'} \mu(s') = \phi(s, a)^\top \hat{\Lambda}^{-1} \left(\sum_{i \in [n]} \phi(s_i, a_i)\right).$$

Thus, if $\phi(s, a)\hat{\Lambda}^{-1}\phi(s_i, a_i) \geq 0$, $\forall (s, a), i$, then it forms a linear MDP with anchor points with $\mathcal{K} = \{(s_i, a_i)\}_{i \in [n]}$, $\lambda_k^{s,a} = \phi(s, a)\hat{\Lambda}^{-1}\phi(s_k, a_k)$ and our analysis can be simply adapted to this case where $nd_m$ is replaced with $\lambda_{\min}(\hat{\Lambda})$ using Freedman's inequality [48], and $\lambda_{\min}(\hat{\Lambda})$ denotes the minimum eigenvalue of $\hat{\Lambda}$. We notice that if $\phi(s, a)\hat{\Lambda}^{-1}\phi(s_i, a_i) < 0$, and $s'_i$ are distinct for each $i$ (which is probable for exponential large $|\mathcal{S}|$), then $\hat{P}(s'_i|s, a) < 0$, which can be pathological for the algorithm and analysis. We expect a well-behaved transition estimation (*i.e.* $\hat{P}$) for linear MDP shares the similar results, even without anchor point assumptions. We leave this as an open problem.

## 7 Conclusion

In this paper, we revisit the offline reinforcement learning on episodic time-homogeneous MDP. We show that, if the total reward is properly normalized, offline reinforcement learning is not harder than the offline bandits counterparts. Specifically, we provide performance guarantee for algorithms based on empirical MDPs, that match the lower bound up to logarithmic factors for offline policy evaluation, and up to logarithmic factors and a higher-order term for offline policy optimization, and both do not have polynomial dependency on $H$. There are still several open problems. For example, can we provide a sharper analysis for the policy obtained by model-based planning without any additional factors on higher-order term? Can we extend to MDP with more general assumption? We leave these problems as future work.

## Acknowledgement

The authors thank the anonymous reviewer for their constructive feedback. TR would like to thank for the helpful discussion with Ming Yin and Yu Bai. SS gratefully acknowledges the funding from NSF grants 1564000 and 1934932, and SSD gratefully acknowledges the funding from NSF Award's IIS-2110170 and DMS-2134106.

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
