# A Proof of the Main Theorems

## A.1 Proof for Offline Policy Evaluation

The proof of all of the technical lemmas can be found in Appendix E. Our proof is organized as follows: we first decompose the estimation error to the errors introduced by reward estimation $\hat{r}$ and transition estimation $\hat{P}$ with Lemma 4, then Lemma 5 provides an upper bound for the error introduced by $\hat{r}$. For error introduced by $\hat{P}$, we first show it can be upper bounded by a "total variance" term (in (9)). A naive bound for this "total variance" term will introduce an additional $H$ factors, thus we apply a recursion-based method to upper bound this "total variance" term (see Lemma 1). Solve the recursion in Lemma 6, and put everything together, we eventually obtain the bound in Theorem 1.

**Lemma 4** (Value Difference Lemma).

$$v^\pi - \hat{v}^\pi = \sum_{h \in [H]} \sum_{s,a} \hat{\xi}_h^\pi(s,a) \left[ r(s,a) - \hat{r}(s,a) + \sum_{s'} \left[ \left( P(s'|s,a) - \hat{P}(s'|s,a) \right) V_{h+1}^\pi(s') \right] \right]. \tag{8}$$

The following lemma provides an upper bound on the estimation error introduced by $\hat{r}$, *i.e.*, the first term in (8).

**Lemma 5** (Error from Reward Estimation). *Suppose Assumption 1 holds, then we have that*

$$\left| \sum_{h \in [H]} \sum_{s,a} \hat{\xi}_h^\pi(s,a) \left[ r(s,a) - \hat{r}(s,a) \right] \right| \leq \sqrt{\frac{\iota}{K d_m}} + \frac{\iota}{K d_m},$$

*holds with probability at least $1 - \delta$.*

We then use a recursive method to bound the error introduced by $\hat{P}$, *i.e.*, the second term in (8). Denote

$$\Delta_1 := \sum_{h \in [H]} \sum_{s,a} \hat{\xi}_h^\pi(s,a) \left[ \sum_{s'} \left( P(s'|s,a) - \hat{P}(s'|s,a) \right) V_{h+1}^\pi(s') \right].$$

With Bernstein's inequality, we know that with high probability

$$\begin{aligned}
\Delta_1 &\leq \sum_{h \in [H]} \sum_{s,a} \hat{\xi}_h^\pi(s,a) \left[ \sqrt{\frac{\mathrm{Var}_{P(s,a)}(V_{h+1}^\pi(s'))\iota}{n(s,a)}} + \frac{\iota}{n(s,a)} \right] \\
&\leq \sqrt{\frac{\iota}{K d_m}} \sqrt{\sum_{h \in [H]} \sum_{s,a} \hat{\xi}_h^\pi(s,a) \mathrm{Var}_{P(s,a)}(V_{h+1}^\pi(s'))} + \frac{\iota}{K d_m},
\end{aligned} \tag{9}$$

where the second inequality is due to Cauchy-Schwartz inequality associated with the Assumption 2.

We then upper bound $\sum_{h \in [H]} \sum_{s,a} \hat{\xi}_h^\pi(s,a) \mathrm{Var}_{P(s,a)}(V_{h+1}^\pi(s'))$ in (9) with Lemma 1. Again, by Bernstein's inequality and Cauchy-Schwarz inequality, we have the following with high probability:

$$\begin{aligned}
&\sum_{h \in [H]} \sum_{s,a} \hat{\xi}_h^\pi(s,a) \left[ \sum_{s'} \left( P(s'|s,a) - \hat{P}(s'|s,a) \right) V_{h+1}^\pi(s')^{2^{i+1}} \right] \\
&\leq \sum_{h \in [H]} \sum_{s,a} \hat{\xi}_h^\pi(s,a) \left[ \sqrt{\frac{\mathrm{Var}_{P(s,a)} \left( V_{h+1}^\pi(s')^{2^{i+1}} \right) \iota}{n(s,a)}} + \frac{\iota}{n(s,a)} \right] \\
&\leq \sqrt{\frac{\iota}{K d_m}} \sqrt{\sum_{h \in [H]} \sum_{s,a} \hat{\xi}_h^\pi(s,a) \mathrm{Var}_{P(s,a)} \left( V_{h+1}^\pi(s')^{2^{i+1}} \right)} + \frac{\iota}{K d_m}.
\end{aligned} \tag{10}$$

Define

$$\mathbb{V}_1(i) := \sum_{h \in [H]} \sum_{s,a} \hat{\xi}_h^\pi(s,a) \mathrm{Var}_{P(s,a)} \left( V_{h+1}^\pi(s')^{2^{i+1}} \right).$$

Apply Lemma 1 with (10), we have the recursion as

$$\mathbb{V}_1(i) \le \sqrt{\frac{\iota}{Kd_m}\mathbb{V}_1(i+1)} + \frac{\iota}{Kd_m} + 2^{i+1}. \tag{11}$$

Notice that $\mathbb{V}_1(i) \le H, \forall i$. Now we can solve the recursion with the following lemma:

**Lemma 6.** *For the recursion formula:*

$$\mathbb{V}(i) \le \sqrt{\lambda_1 \mathbb{V}(i+1)} + \lambda_1 + 2^{i+1}\lambda_2,$$

*with $\lambda_1, \lambda_2 > 0$, if $\mathbb{V}(i) \le H, \forall i$, then we have that*

$$\mathbb{V}(0) \le 6(\lambda_1 + \lambda_2),$$

*and we need to do the recursion at most $O(\log \frac{H\lambda_1}{\lambda_2^2})$ times.*

**Remark** We want to emphasize that, the recursion needs to be done at most $O(\log \frac{H\lambda_1}{\lambda_2})$ times. Thus, by union bound, such recursion only introduces an additional $\log \log$ factor in the error when $\lambda_1, \lambda_2 = \text{poly}(S, A, H)$, that can be absorbed by $\iota$. For simplicity, we still use $\iota$ to denote the poly log factors in the following derivation.

Apply Lemma 6 with $\lambda_1 = \frac{\iota}{Kd_m}$, $\lambda_2 = 1$, we have that

$$\mathbb{V}_1(0) = O\left(\frac{\iota}{Kd_m} + 1\right),$$

and also notice in (9) that

$$\Delta_1 \le \sqrt{\frac{\iota}{Kd_m}\mathbb{V}_1(0)} + \frac{\iota}{Kd_m}.$$

Combine this two inequality, we have that

$$\Delta_1 \le O\left(\sqrt{\frac{\iota}{Kd_m}\left(\frac{\iota}{Kd_m} + 1\right)}\right) + \frac{\iota}{Kd_m}.$$

Suppose $K = \widetilde{\Omega}\left(\frac{1}{d_m}\right)$, we have that

$$\Delta_1 \le O\left(\sqrt{\frac{\iota}{Kd_m}} + \frac{\iota}{Kd_m}\right). \tag{12}$$

Combined (12) with Lemma 5 and $K = \widetilde{\Omega}\left(\frac{1}{d_m}\right)$, we conclude the proof of Theorem 1.

## A.2 Proof for Offline Policy Optimization

We first make the following standard decomposition:

$$v^{\pi^*} - v^{\hat{\pi}^*} = v^{\pi^*} - \hat{v}^{\pi^*} + \underbrace{\hat{v}^{\pi^*} - \hat{v}^{\hat{\pi}^*}}_{\le 0} + \hat{v}^{\hat{\pi}^*} - v^{\hat{\pi}^*}$$

$$\le \underbrace{v^{\pi^*} - \hat{v}^{\pi^*}}_{\text{Error on Fixed Policy}} + \underbrace{\hat{v}^{\hat{\pi}^*} - v^{\hat{\pi}^*}}_{\text{Error on Data−Dependent Policy}}. \tag{13}$$

The first term characterizes the evaluation difference of optimal policy on original MDP and the empirical MDP, and the second term characterize the evaluation difference of the planning result $\hat{\pi}^*$ from the empirical MDP on original MDP and the empirical MDP.

We can directly apply Theorem 1 to bound the first term in (13). However, as $\hat{\pi}^*$ has complicated statistical dependency with $\hat{P}$, we cannot apply Theorem 1 on the second term in (13) for the evaluation error on data-dependent policy. Notice that a direct application of the absorbing MDP techniques introduced in [34, 28] for the second term will introduce additional $H$ or $S$ factors in the main term as shown in [1]. Thus, we further generalize our recursion-based method to keep the main

term tight while only introduce an additional $S$ factor at the higher-order term, which keeps the final error horizon-free.

Similar to the case in the offline evaluation, we make the following decomposition based on Lemma 4:

$$\hat{v}^{\hat{\pi}^*} - v^{\hat{\pi}^*} = \sum_{h \in [H]} \sum_{s,a} \hat{\xi}_h^{\hat{\pi}^*}(s,a) \left[ (\hat{r}(s,a) - r(s,a)) + \sum_{s'} (\hat{P}(s'|s,a) - P(s'|s,a)) V_{h+1}^{\hat{\pi}^*}(s') \right]$$

$$= \sum_{h \in [H]} \sum_{s,a} \hat{\xi}_h^{\hat{\pi}^*}(s,a) \left( \Delta_r(s,a) + \Delta_P(s,a) + \Delta_{PV}(s,a) \right),$$

where

$$\begin{aligned} \Delta_r(s,a) &:= \hat{r}(s,a) - r(s,a), \\ \Delta_P(s,a) &:= \sum_{s'} (\hat{P}(s'|s,a) - P(s'|s,a)) V_{h+1}^{\pi^*}(s') \\ \Delta_{PV}(s,a) &:= \sum_{s'} \left( \hat{P}(s'|s,a) - P(s'|s,a) \right) \left( V_{h+1}^{\hat{\pi}^*}(s') - V_{h+1}^{\pi^*}(s') \right). \end{aligned}$$

For the inner product of $\hat{\xi}_h^{\hat{\pi}^*}(s,a)$ with $\Delta_r$, as $r$ is independent of $\hat{P}$, we can identically apply the result for offline evaluation, that leads to a $\tilde{O}\left( \sqrt{\frac{1}{Kd_m}} + \frac{1}{Kd_m} \right)$ error. We then consider the error induced by $\Delta_P$ and $\Delta_{PV}$.

For the error introduced by $\Delta_P$, as $\pi^*$ is independent of $\hat{P}$, we can use Bernstein inequality and Cauchy-Schwartz inequality and obtain

$$\Delta_2 := \left| \sum_{h \in [H]} \sum_{s,a} \hat{\xi}_h^{\pi}(s,a) \Delta_P(s,a) \right|$$

$$\leq \sum_{h \in [H]} \sum_{s,a} \hat{\xi}_h^{\pi}(s,a) \cdot \left[ \sqrt{\frac{\cdot \text{Var}_{P(s,a)} \left( V_{h+1}^{\pi^*}(s') \right) \iota}{n(s,a)}} + \frac{\iota}{n(s,a)} \right]$$

$$\leq \sqrt{\frac{\iota}{Kd_m}} \sqrt{\sum_{h \in [H]} \sum_{s,a} \hat{\xi}_h^{\pi}(s,a) \text{Var}_{P(s,a)} \left( V_{h+1}^{\pi^*}(s') \right)} + \frac{\iota}{Kd_m}. \tag{14}$$

Now we turn to $\sum_{h \in [H]} \sum_{s,a} \hat{\xi}_h^{\pi}(s,a) \text{Var}_{P(s,a)} \left( V_{h+1}^{\pi^*}(s') \right)$. With Lemma 3, we have that

$$\sum_{h \in [H]} \sum_{s,a} \hat{\xi}_h^{\pi}(s,a) \text{Var}_{P(s,a)} \left( (V_{h+1}^{\pi^*}(s'))^{2^i} \right)$$

$$\leq \sum_{h \in [H]} \sum_{s,a} \hat{\xi}_h^{\pi}(s,a) \left[ \sum_{s'} \left( P(s'|s,a) - \hat{P}(s'|s,a) \right) \left( V_{h+1}^{\pi^*}(s') \right)^{2^{i+1}} \right] + 2^{i+1} \left( \Delta_2 + v^{\pi^*} \right). \tag{15}$$

We can further apply Bernstein inequality and Cauchy-Schwartz inequality to the first term in (15). Specifically, denote

$$\mathbb{V}_2(i) := \sum_{h \in [H]} \sum_{s,a} \hat{\xi}_h^{\pi}(s,a) \text{Var}_{P(s,a)} \left( \left( V_{h+1}^{\pi^*}(s') \right)^{2^i} \right),$$

we have the recursion as

$$\mathbb{V}_2(i) \leq \sqrt{\frac{\iota}{Kd_m} \mathbb{V}_2(i+1)} + \frac{\iota}{Kd_m} + 2^{i+1} \left( \Delta_2 + v^{\pi^*} \right).$$

This recursion can be solved similarly as (11) by applying Lemma 6 with $\lambda_1 = \frac{\iota}{Kd_m}$ and $\lambda_2 = \Delta_2 + v^{\pi^*}$, which leads to

$$\mathbb{V}_2(0) \leq O\left( \frac{\iota}{Kd_m} + \left( v^{\pi^*} + \Delta_2 \right) \right). \tag{16}$$

Meanwhile, from (14) we have that

$$\Delta_2 \le \sqrt{\frac{\iota}{Kd_m}\mathbb{V}_2(0)} + \frac{\iota}{Kd_m}. \tag{17}$$

Combine (16) and (17), we have

$$\Delta_2 \le O\left(\sqrt{\frac{\iota}{Kd_m}\left(\frac{\iota}{Kd_m} + v^{\pi^*} + \Delta_2\right)}\right) + \frac{\iota}{Kd_m},$$

Suppose $K = \tilde{\Omega}\left(\frac{1}{d_m}\right)$, then with Assumption 1 and the discussion in D, we have that

$$\Delta_2 \le O\left(\sqrt{\frac{\iota}{Kd_m}} + \frac{\iota}{Kd_m}\right). \tag{18}$$

Now we turn to the error introduced by $\Delta_{PV}$. With Lemma 2, we can again use the recursion to bound it, and finally obtain the bound in Theorem 3. By Lemma 2 and Cauchy-Schwartz inequality, we have that

$$\Delta_3 := \left|\sum_{h\in[H]}\sum_{s,a}\hat{\xi}_h^\pi(s,a)\Delta_{PV}(s,a)\right|$$

$$\le \sum_{h\in[H]}\sum_{s,a}\hat{\xi}_h^\pi(s,a)\cdot\left[\sqrt{\frac{S\cdot\mathrm{Var}_{P(s,a)}(V_{h+1}^{\pi^*}(s') - V_{h+1}^{\hat{\pi}^*}(s'))\iota}{n(s,a)}} + \frac{S\iota}{n(s,a)}\right]$$

$$\le \sqrt{\frac{S\iota}{Kd_m}}\sqrt{\sum_{h\in[H]}\sum_{s,a}\hat{\xi}_h^\pi(s,a)\mathrm{Var}_{P(s,a)}\left(V_{h+1}^{\pi^*}(s') - V_{h+1}^{\hat{\pi}^*}(s')\right)} + \frac{S\iota}{Kd_m}. \tag{19}$$

Now we turn to $\sum_{h\in[H]}\sum_{s,a}\hat{\xi}_h^\pi(s,a)\mathrm{Var}_{P(s,a)}(V_{h+1}^{\pi^*}(s') - V_{h+1}^{\hat{\pi}^*}(s'))$ in (19). We still bound this term with the recursive methods. With Lemma 3, we have that

$$\sum_{h\in[H]}\sum_{s,a}\hat{\xi}_h^\pi(s,a)\mathrm{Var}_{P(s,a)}\left((V_{h+1}^{\pi^*}(s') - V_{h+1}^{\hat{\pi}^*}(s'))^{2^i}\right)$$

$$\le \sum_{h\in[H]}\sum_{s,a}\hat{\xi}_h^\pi(s,a)\left[\sum_{s'}\left(P(s'|s,a) - \hat{P}(s'|s,a)\right)\left(V_{h+1}^{\pi^*}(s') - V_{h+1}^{\hat{\pi}^*}(s')\right)^{2^{i+1}}\right] + 2^{i+1}\left(\Delta_3 + \left(v^{\pi^*} - v^{\hat{\pi}^*}\right)\right). \tag{20}$$

We can further apply Lemma 2 and Cauchy-Schwartz inequality to the first term in (20), which eventually lead to the recursion formula. Specifically, denote

$$\mathbb{V}_3(i) := \sum_{h\in[H]}\sum_{s,a}\hat{\xi}_h^\pi(s,a)\mathrm{Var}_{P(s,a)}\left(\left(V_{h+1}^{\pi^*}(s') - V_{h+1}^{\hat{\pi}^*}(s')\right)^{2^i}\right),$$

we have the recursion as

$$\mathbb{V}_3(i) \le \sqrt{\frac{S\iota}{Kd_m}\mathbb{V}_3(i+1)} + \frac{S\iota}{Kd_m} + 2^{i+1}\left(\Delta_3 + v^{\pi^*} - v^{\hat{\pi}^*}\right).$$

This recursion can be solved similarly as (11) by applying Lemma 6 with $\lambda_1 = \frac{S\iota}{Kd_m}$ and $\lambda_2 = \Delta_3 + v^{\pi^*} - v^{\hat{\pi}^*}$, which leads to

$$\mathbb{V}_3(0) \le O\left(\frac{S\iota}{Kd_m} + \left(v^{\pi^*} - v^{\hat{\pi}^*} + \Delta_3\right)\right). \tag{21}$$

Meanwhile, from (19) we have that

$$\Delta_3 \le \sqrt{\frac{S\iota}{Kd_m}\mathbb{V}_3(0)} + \frac{S\iota}{Kd_m}. \tag{22}$$

Combine (21) and (22), we have

$$\Delta_3 \leq O\left(\sqrt{\frac{S\iota}{Kd_m}\left(\frac{S\iota}{Kd_m} + v^{\pi^*} - v^{\hat{\pi}^*} + \Delta_3\right)}\right) + \frac{S\iota}{Kd_m},$$

Moreover, with (13) and (18), we have that

$$v^{\pi^*} - v^{\hat{\pi}^*} \leq O\left(\sqrt{\frac{\iota}{Kd_m}} + \frac{\iota}{Kd_m}\right) + \Delta_3.$$

Thus, with the discussion in Appendix D, we can conclude that $v^{\pi^*} - v^{\hat{\pi}^*} \leq O\left(\sqrt{\frac{\iota}{Kd_m}} + \frac{S\iota}{Kd_m}\right)$, which finishes the proof for Theorem 3.

# B  Proof of the Lower Bounds

## B.1  Lower Bound for Offline Policy Evaluation

Our lower bound instance is adapted from the instances in [35, 49, 29] for finite horizon time-homogeneous setting.

*Proof.* We consider a two-state MDP with state $s_1, s_2$, with an unique action $a$. $s_1$ is an absorbing state, which means $P(s_1|s_1, a) = 1$, while $P(s_2|s_2, a) = p$, $P(s_1|s_2, a) = 1 - p$. We assume the reward is deterministic with $r(s_1, a) = 0$, $r(s_2, a) = \frac{1}{H}$ that satisfies Assumption 1. Assume we want to have an accurate estimation of $V_1(s_2)$, which is equivalent to have a sufficient accurate estimation of $p$. With straightforward calculation, we have that

$$V(p) := V_1(s_2) = \frac{p - p^{H+1}}{1 - p}\frac{1}{H}.$$

Notice that

$$\frac{\partial V(p)}{\partial p} = \frac{1 - (1 + (1 - p)H)p^H}{(1 - p)^2}\frac{1}{H}.$$

Let $p_1 = 1 - \frac{c_1}{H}$, where $c_1$ is an absolute constant, we know that

$$\frac{\partial V(p_1)}{\partial p_1} = \frac{1 - c_1(1 - \frac{c_1}{H})^H}{(1 - p_1)^2}\frac{1}{H} \geq \frac{1 - c_1 e^{-c_1}}{(1 - p_1)^2}\frac{1}{H} = \frac{1 - c_1 e^{-c_1}}{c_1^2}H,$$

which is monotonically decreasing w.r.t $c_1$. Assume $p_2 = 1 - \frac{c_2}{H}$ where $c_2 < c_1$ is another absolute constant, we have that

$$V(p_2) - V(p_1) \geq \frac{1 - c_1 e^{-c_1}}{(1 - p_1)^2}\frac{1}{H}(p_2 - p_1) = \frac{1 - c_1 e^{-c_1}}{c_1^2}(c_1 - c_2).$$

We now use Le Cam's method to show that without sufficient number of data from $\text{Bern}(p)$, we cannot identify $p = p_1$ or $p = p_2$ with high probability, and thus cannot have ideal estimation error on both of $p_1$ and $p_2$. We start from the following lemma:

**Lemma 7.**

$$KL(Bern(p)\|Bern(q)) \leq \frac{(p - q)^2}{q(1 - q)}.$$

*Proof.*

$$\begin{aligned}
&\text{KL}(\text{Bern}(p)\|\text{Bern}(q)) \\
=&p \log \frac{p}{q} + (1 - p)\log\frac{1 - p}{1 - q} \\
\leq&p\frac{p - q}{q} + (1 - p)\log\frac{q - p}{1 - p} \\
=&\frac{(p - q)^2}{q(1 - q)},
\end{aligned}$$

where the inequality is due to the fact that $\log(1 + x) \leq x$. □

Assume $\Psi : [0,1]^{n(s_2,a)} \to \{p_1, p_2\}$ is a test with $n$ i.i.d samples from $\mathrm{Bern}(p)$, and use $\mathbb{P}_1$ and $\mathbb{P}_2$ to denote the probability measure under $p_1$ and $p_2$, we have that

$$
\begin{aligned}
&\inf_{\Psi} \left\{ \mathbb{P}_1(\Psi(\mathcal{D}) \neq p_1) + \mathbb{P}_2(\Psi(\mathcal{D}) \neq p_2) \right\} \\
&\geq 1 - \|(\mathrm{Bern}(p_1))^{n(s_2,a)} - (\mathrm{Bern}(p_2))^{n}\|_{\mathrm{TV}} \quad \text{(Le Cam's inequality)} \\
&\geq 1 - \sqrt{\frac{n(s_2,a)}{2} \mathrm{KL}\left(\mathrm{Bern}(p_1)\|\mathrm{Bern}(p_2)\right)} \quad \text{(Pinsker's inequality)} \\
&\geq 1 - \sqrt{\frac{n(s_2,a)(p_1 - p_2)^2}{p_1(1-p_1)}} \\
&= 1 - \sqrt{\frac{n(s_2,a)(c_1 - c_2)^2}{c_1(H - c_1)}}.
\end{aligned}
$$

Take $c_2 = c_1 - \sqrt{\frac{c_1(H-c_1)}{2n(s_2,a)}}$, we know that with probability at least $0.5$ we cannot identify $p = p_1$ or $p = p_2$. And notice that

$$
V(p_2) - V(p_1) \geq \frac{1 - c_1 e^{-c_1}}{c_1^2} \sqrt{\frac{c_1(H - c_1)}{2n(s_2,a)}} \geq 2c_0 \sqrt{\frac{H}{n(s_2,a)}},
$$

where $c_0$ is an absolute constant only depends on $c_1$. Thus we know that, with $n(s_2,a)$ samples from $P(s_2, a)$, we must suffer from an estimation error of $\Omega\left(\sqrt{\frac{H}{n(s_2,a)}}\right)$ with probability at least $0.25$. Notice that we can set $n(s_2,a) = nd_m$, thus finishes the proof. $\square$

### B.2 Lower Bound of Offline Policy Improvement

We can further show the lower bound of offline improvement for finite horizon time-homogeneous MDP, based on the hard instance we mentioned above.

*Proof.* We introduce additional states $s_0$ and $s_3$ in the previous hard instance, with the transition from $s_0$, $P(s_1|s_0, a_1) = 1$, $P(s_3|s_0, a) = 1$, $\forall a \neq a_1$, and $s_3$ is an absorbing state with total reward in $H$ steps (i.e. $V_1(s_3)$) as $V(p_1) + c_0\sqrt{\frac{H}{nd_m}}$. We always start from $s_0$, and we need to choose the action at $s_0$. Notice that, if $p = p_2$, then the optimal arm is $a_1$, while if $p = p_1$, then the optimal arm is not $a_1$, both with a sub-optimal gap of at least $c_0\sqrt{\frac{H}{nd_m}}$, which finishes the proof. $\square$

## C Proof for Linear MDP with Anchor Points

### C.1 Proof for Offline Policy Evaluation

Notice that, the value difference lemma holds for any kinds of MDP. Thus, if we have Bernstein-type concentration for $\hat{r}(s,a) - r(s,a)$ and $\sum_{s'}(P(s'|s,a) - \hat{P}(s'|s,a))V(s')$, we can adopt the techniques for tabular MDP and obtain the desired results.

**Lemma 8.** *With probability at least $1 - \delta$, we have that $\forall (s,a)$,*

$$
|\hat{r}(s,a) - r(s,a)| \leq \sqrt{\frac{r(s,a)}{nd_m}} + \frac{\iota}{nd_m}
$$

*where $\iota$ absorbs the logarithm factors $\log(\mathrm{poly}(d)/\delta)$.*

*Proof.* Notice that

$$|\hat{r}(s,a) - r(s,a)|$$

$$\leq \sum_{k\in\mathcal{K}} \lambda_k^{s,a} |\hat{r}(s_k,a_k) - r(s_k,a_k)|$$

$$\leq \sum_{k\in\mathcal{K}} \lambda_k^{s,a} \left( \sqrt{\frac{r(s_k,a_k)}{nd_m}} + \frac{1}{nd_m} \right)$$

$$\leq \sqrt{\frac{\sum_{k\in\mathcal{K}} \lambda_k^{s,a} r(s_k,a_k)}{nd_m}} + \frac{\iota}{nd_m}$$

$$= \sqrt{\frac{r(s,a)}{nd_m}} + \frac{\iota}{nd_m},$$

where the second inequality is due to the Bernstein's inequality on each $(s_k,a_k)$ with $\iota = \log(2d/\delta)$ and $r(s_k,a_k) \in [0,1]$, and the third inequality is due to Cauchy-Schwartz inequality and $\sum_{k\in\mathcal{K}} \lambda_k^{s,a} = 1$ $\qquad\square$

**Lemma 9.** *Suppose $V(s')$ is independent from $\hat{P}(s'|s,a)$, then with probability at least $1 - \delta$, we have that $\forall(s,a)$*

$$\left| \sum_{s'} (P(s'|s,a) - \hat{P}(s'|s,a))V(s') \right| \leq \sqrt{\frac{Var_{P(s,a)}V(s')\iota}{nd_m}} + \frac{\iota}{nd_m},$$

*where $\iota$ absorbs the logarithm factors $\log(\mathrm{poly}(d)/\delta)$.*

*Proof.* First, we have that

$$\left| \sum_{s'} (P(s'|s,a) - \hat{P}(s'|s,a))V(s') \right|$$

$$\leq \sum_{k\in\mathcal{K}} \lambda_k^{s,a} \left| \sum_{s'} \left( P(s'|s_k,a_k) - \hat{P}(s'|s_k,a_k) \right) V(s') \right|$$

$$\leq \sum_{k\in\mathcal{K}} \lambda_k^{s,a} \left( \sqrt{\frac{\mathrm{Var}_{P(s_k,a_k)}V(s')\iota}{nd_m}} + \frac{\iota}{nd_m} \right)$$

$$\leq \sqrt{\frac{\sum_{k\in\mathcal{K}} \lambda_k^{s,a} \mathrm{Var}_{P(s_k,a_k)}V(s')\iota}{nd_m}} + \frac{\iota}{nd_m},$$

where the second inequality is due to the Bernstein's inequality on each $(s_k,a_k)$ with $\iota = \log(2d/\delta)$ and the last inequality is due to Cauchy-Schwartz inequality and $\sum_{k\in\mathcal{K}} \lambda_k^{s,a} = 1$. Notice that

$$\sum_{k\in\mathcal{K}} \lambda_k^{s,a} \mathrm{Var}_{P(s_k,a_k)}V(s')$$

$$= \sum_{k\in\mathcal{K}} \lambda_k^{s,a} \left( \sum_{s'} P(s'|s_k,a_k)V(s')^2 - \left( \sum_{s'} P(s'|s_k,a_k)V(s') \right)^2 \right)$$

$$\leq \sum_{s'} P(s'|s,a)V(s')^2 - \left( \sum_{k\in\mathcal{K}} \lambda_k^{s,a} P(s'|s_k,a_k)V(s') \right)^2$$

$$= \mathrm{Var}_{P(s,a)}V(s'),$$

where the inequality is due to Cauchy-Schwartz inequality and $\sum_{k\in\mathcal{K}} \lambda_k^{s,a} = 1$. Substitute this term back and we conclude the proof. $\qquad\square$

Notice that, Lemma 9 simultaneously holds for all of the $(s, a)$ if all of the concentration on the anchor points hold. Hence we can apply the analysis for tabular MDP and obtain the desired results, which finishes the proof for offline policy evaluation on linear MDP with anchor points.

## C.2 Proof for Offline Policy Optimization

Here we need a Bernstein-type concentration for $\sum_{s'}(P(s'|s,a) - \hat{P}(s'|s,a))V(s')$ when $V(s')$ and $\hat{P}(s'|s,a)$ are correlated. A naive application of Lemma 2 will introduce a $|\mathcal{S}|$ factor in the higher order term, which is not satisfactory, as $|S|$ can be exponentially large. Here use another method to replace this dependency on $|\mathcal{S}|$ with the feature dimension $d$.

**Lemma 10.** *Suppose $\tilde{V}(s')$ is independent from $\hat{P}(s'|s,a)$, then with probability $1 - \delta$, we have that*

$$\left| \sum_{s'} (P(s'|s,a) - \hat{P}(s'|s,a))V(s') \right| \le \sqrt{\frac{Var_{P(s,a)}V(s')\iota}{nd_m}} + \frac{\iota}{nd_m} + \|\tilde{V} - V\|_\infty \left( 1 + \sqrt{\frac{\iota}{nd_m}} \right),$$

*where $\iota$ absorbs the logarithm factors $\log(\text{poly}(d)/\delta)$.*

*Proof.* Notice that

$$\left| \sum_{s'} (P(s'|s,a) - \hat{P}(s'|s,a))V(s') \right|$$

$$\le \left| \sum_{s'} (P(s'|s,a) - \hat{P}(s'|s,a))\tilde{V}(s') \right| + \left| \sum_{s'} (P(s'|s,a) - \hat{P}(s'|s,a))(V(s') - \tilde{V}(s')) \right|$$

$$\le \sqrt{\frac{\text{Var}_{P(s,a)}\tilde{V}(s')\iota}{nd_m}} + \frac{\iota}{nd_m} + \|\tilde{V} - V\|_\infty$$

$$\le \sqrt{\frac{\text{Var}_{P(s,a)}V(s')\iota}{nd_m}} + \sqrt{\frac{\text{Var}_{P(s,a)}(\tilde{V}(s') - V(s'))\iota}{nd_m}} \frac{\iota}{nd_m} + \|\tilde{V} - V\|_\infty$$

$$\le \sqrt{\frac{\text{Var}_{P(s,a)}\tilde{V}(s')\iota}{nd_m}} + \frac{\iota}{nd_m} + \|\tilde{V} - V\|_\infty \left( 1 + \sqrt{\frac{\iota}{nd_m}} \right),$$

where the first inequality is due to the triangle inequality, the second inequality is due to Lemma 9 and algebra, the third inequality is due to the triangle inequality for the variance, *i.e.* $\sqrt{\text{Var}(X + Y)} \le \sqrt{\text{Var}(X)} + \sqrt{\text{Var}(Y)}$, and the last inequality is due to the fact that $\text{Var}(V) \le \|V\|_\infty$. □

With Lemma 10, we can construct an $\epsilon$-net (under $\ell_\infty$ norm) for $V$ to obtain Bernstein-type concentration. For tabular MDP, this $\epsilon$-net is of size $O(\epsilon^{-|\mathcal{S}|})$, which leads to the same result of Lemma 2. However, in linear MDP, $Q$ follows a linear form $Q(s,a) = \phi(s,a)^\top w_Q$, thus the $V$ we consider lies in a $d$-dimensional manifolds, and the size of the $\epsilon$-net we exactly need is $O(\epsilon^{-d})$. This observation leads to the following corollary:

**Corollary 1.** *For any $V(s)'$, with probability $1 - \delta$, we have that*

$$\left| \sum_{s'} (P(s'|s,a) - \hat{P}(s'|s,a))V(s') \right| \le \sqrt{\frac{d Var_{P(s,a)}V(s')\iota}{nd_m}} + \frac{d\iota}{nd_m} + \epsilon \left( 1 + \sqrt{\frac{d\iota}{nd_m}} \right),$$

*where $\iota$ absorbs the logarithm factors, $\log(\text{poly}(d, 1/\epsilon)/\delta)$.*

Here the additional $d$ comes from the logarithm of the size of $\epsilon$-net. We can further choose $\epsilon = \frac{d}{nd_m}$ and absorb $\epsilon \left( 1 + \sqrt{\frac{d\iota}{nd_m}} \right)$ into $\frac{d\iota}{nd_m}$, then apply the analysis for tabular MDP and replace $S$ with $d$ to obtain the desired result, hence conclude the proof.

# D Step-by-Step Solving for $\Delta_2$ and $v^{\pi^*} - v^{\hat{\pi}^*}$

## D.1 Explicit Bound for $\Delta_2$

Notice that, for some absolute constant $c$,

$$\Delta_2 \leq c\sqrt{\frac{\iota}{Kd_m}\left(\frac{\iota}{Kd_m} + \Delta_2 + v^{\pi^*}\right)} + \frac{\iota}{Kd_m}.$$

which means

$$c\Delta_2 \leq \sqrt{\frac{c\iota}{Kd_m}\left(\frac{c\iota}{Kd_m} + c\Delta_2 + cv^{\pi^*}\right)} + \frac{c\iota}{Kd_m}$$

$$\leq \frac{c\Delta_2}{2} + \frac{cv^{\pi^*}}{2} + \frac{2c\iota}{Kd_m},$$

thus we have that

$$\Delta_2 \leq v^{\pi^*} + \frac{4\iota}{Kd_m}.$$

Substitute back, suppose $N = \widetilde{\Omega}\left(\frac{1}{d_m}\right)$, then with Assumption 1, we have that

$$\Delta_2 \leq c\sqrt{\frac{\iota}{Kd_m}\left(\frac{5\iota}{Kd_m} + 2v^{\pi^*}\right)} + \frac{\iota}{Kd_m} = O\left(\sqrt{\frac{\iota}{Kd_m}} + \frac{\iota}{Kd_m}\right).$$

## D.2 Explicit bound for $v^{\pi^*} - v^{\hat{\pi}^*}$

Notice that, for some absolute constant $c$, we have that

$$\Delta_3 \leq c\sqrt{\frac{S\iota}{Kd_m}\left(\frac{S\iota}{Kd_m} + v^{\pi^*} - v^{\hat{\pi}^*} + \Delta_3\right)} + \frac{S\iota}{Kd_m},$$

which means

$$c\Delta_3 \leq \sqrt{\frac{cS\iota}{Kd_m}\left(\frac{cS\iota}{Kd_m} + c(v^{\pi^*} - v^{\hat{\pi}^*}) + c\Delta_3\right)} + \frac{cS\iota}{Kd_m}$$

$$\leq \frac{c(v^{\pi^*} - v^{\hat{\pi}^*})}{2} + \frac{c\Delta_3}{2} + \frac{2cS\iota}{Kd_m},$$

thus we have that

$$\Delta_3 \leq (v^{\pi^*} - v^{\hat{\pi}^*}) + \frac{4S\iota}{Kd_m}.$$

We then substitute back, and know that

$$\Delta_3 \leq c\sqrt{\frac{S\iota}{Kd_m}\left(2(v^{\pi^*} - v^{\hat{\pi}^*}) + \frac{5S\iota}{Kd_m}\right)} + \frac{S\iota}{Kd_m}.$$

Furthermore, for another absolute constant $c'$, we have that

$$v^{\pi^*} - v^{\hat{\pi}^*} \leq c'\left(\sqrt{\frac{\iota}{Kd_m}} + \frac{\iota}{Kd_m}\right) + \Delta_3$$

$$\leq c'\sqrt{\frac{\iota}{Kd_m}} + c\sqrt{\frac{S\iota}{Kd_m}\left(2(v^{\pi^*} - v^{\hat{\pi}^*}) + \frac{5S\iota}{Kd_m}\right)} + \frac{(c'+1)S\iota}{Kd_m},$$

which means

$$\left(\sqrt{2(v^{\pi^*} - v^{\hat{\pi}^*}) + \frac{5S\iota}{Kd_m}} - c\sqrt{\frac{S\iota}{Kd_m}}\right)^2 \leq 2c'\sqrt{\frac{\iota}{Kd_m}} + \frac{(c^2 + 2c' + 2)S\iota}{Kd_m},$$

that can be translated to the bound

$$v^{\pi^*} - v^{\hat{\pi}^*} \leq \left( \sqrt{2c'\sqrt{\frac{\iota}{Kd_m}} + \frac{(c^2 + 2c' + 2)S\iota}{Kd_m}} + c\sqrt{\frac{S\iota}{Kd_m}} \right)^2 - \frac{5S\iota}{Kd_m}$$

$$\leq 2 \left( \sqrt{2c'\sqrt{\frac{\iota}{Kd_m}} + \frac{(2c^2 + 2c' + 2)S\iota}{Kd_m}} \right)^2$$

$$= O\left( \sqrt{\frac{\iota}{Kd_m}} + \frac{S\iota}{Kd_m} \right)$$

where we use $\sqrt{a} + \sqrt{b} \leq \sqrt{2(a+b)}$.

# E   Proof of Technical Lemmas

## E.1   Bernstein's Inequality

**Lemma 11** (Bernstein's Inequality). *Let $\{X_i\}_{i=1}^n$ be i.i.d random variables from $X$ with values bounded in $[0, 1]$, then with probability at least $1 - \delta$, we have that*

$$\left| \sum_{i=1}^n X_i - \mathbb{E}[X] \right| \leq \sqrt{\frac{2Var(X)\log\frac{2}{\delta}}{n}} + \frac{\log\frac{2}{\delta}}{3n},$$

*where $Var(X)$ is the variance of $X$.*

For the proof of Bernstein's inequality, we refer the interested reader to Wainwright [50].

### E.2 Proof of Lemma 1

*Proof.* We have that

$$\sum_{h \in [H]} \sum_{s,a} \hat{\xi}_h^\pi(s,a) \text{Var}_{P(s,a)}\left(V_{h+1}^\pi(s')^{2^i}\right)$$

$$= \sum_{h \in [H]} \sum_{s,a} \hat{\xi}_h^\pi(s,a)\left[\sum_{s'} P(s'|s,a)V_{h+1}^\pi(s')^{2^{i+1}} - \left(\sum_{s'} P(s'|s,a)V_{h+1}^\pi(s')^{2^i}\right)^2\right]$$

$$= \sum_{h \in [H]} \sum_{s,a} \hat{\xi}_h^\pi(s,a)\left[\sum_{s'} \left(P(s'|s,a) - \hat{P}(s'|s,a)\right)V_{h+1}^\pi(s')^{2^{i+1}} + V_h^\pi(s)^{2^{i+1}} - \left(\sum_{s'} P(s'|s,a)V_{h+1}^\pi(s')^{2^i}\right)^2\right]$$

$$- \sum_s \mu(s)V_1^\pi(s)^{2^{i+1}}$$

$$\leq \sum_{h \in [H]} \sum_{s,a} \hat{\xi}_h^\pi(s,a)\left[\sum_{s'} \left(P(s'|s,a) - \hat{P}(s'|s,a)\right)V_{h+1}^\pi(s')^{2^{i+1}}\right]$$

$$+ \sum_{h \in [H]} \sum_{s,a} \hat{\xi}_h^\pi(s,a)\left[Q_h^\pi(s,a)^{2^{i+1}} - \left(\sum_{s'} P(s'|s,a)V_{h+1}^\pi(s')\right)^{2^{i+1}}\right]$$

$$\leq \sum_{h \in [H]} \sum_{s,a} \hat{\xi}_h^\pi(s,a)\left[\sum_{s'} \left(P(s'|s,a) - \hat{P}(s'|s,a)\right)V_{h+1}^\pi(s')^{2^{i+1}}\right]$$

$$+ 2^{i+1} \sum_{h \in [H]} \sum_{s,a} \hat{\xi}_h^\pi(s,a)\left[Q_h^\pi(s,a) - \left(\sum_{s'} P(s'|s,a)V_{h+1}^\pi(s')\right)\right],$$

$$= \sum_{h \in [H]} \sum_{s,a} \hat{\xi}_h^\pi(s,a)\left[\sum_{s'} \left(P(s'|s,a) - \hat{P}(s'|s,a)\right)V_{h+1}^\pi(s')^{2^{i+1}}\right] + 2^{i+1} \sum_{h \in [H]} \sum_{s,a} \hat{\xi}_h^\pi(s,a)r(s,a)$$

where in the second step we use the fact that

$$\sum_{s,a} \hat{\xi}_h^\pi(s,a)\left[\sum_{s'} \hat{P}(s'|s,a)V_{h+1}(s')^{2^{i+1}}\right]$$

$$= \sum_{s'}\left[\sum_{s,a} \hat{\xi}_h^\pi(s,a)\hat{P}(s'|s,a)\right]V_{h+1}(s')^{2^{i+1}}$$

$$= \sum_{s'} \hat{\xi}_{h+1}^\pi(s')V_{h+1}(s')^{2^{i+1}}$$

$$= \sum_{s',a'} \hat{\xi}_{h+1}^\pi(s',a')V_{h+1}(s')^{2^{i+1}}.$$

We drop the $\sum_s \mu(s)V_1^\pi(s)^{2^{i+1}}$ and use the convexity of $x^{2^i}$ and $V_h^\pi(s) = \mathbb{E}_\pi Q_h^\pi(s,a)$ in the third step, and the last step is indicated by the assumption that $V_h(s) \leq 1, \forall h \in [H], s \in \mathcal{S}$.

With Assumption 1, we know that

$$\sum_{h \in [H]} \sum_{s,a} \hat{\xi}_h^\pi(s,a)r(s,a) \leq 1,$$

as each trajectory that can generated by $\widehat{\mathcal{M}}$ can be generated by $\mathcal{M}$, thus finish the proof. □

## E.3 Proof of Lemma 2

*Proof.*

$$\left| \sum_{s'} \Big( P(s'|s,a) - \hat{P}(s'|s,a) \Big) V_h(s') \right|$$

$$= \left| \sum_{s'} \Big( P(s'|s,a) - \hat{P}(s'|s,a) \Big) \left( V_h(s') - \sum_{s'} P(s'|s,a) V_h(s') \right) \right|$$

$$\leq \sum_{s'} \sqrt{\frac{P(s'|s,a)\iota}{n(s,a)}} \left| V_h(s') - \sum_{s'} P(s'|s,a) V_h(s') \right| + \frac{S\iota}{n(s,a)}$$

$$\leq \sqrt{\frac{S\mathrm{Var}_{P(s,a)}\left(V_h(s')\right)\iota}{n(s,a)}} + \frac{S\iota}{n(s,a)},$$

where the first equality is due to the fact that $\sum_{s'} P(s'|s,a) = \sum_{s'} \hat{P}(s'|s,a) = 1$, the second inequality is due to Bernstein's inequality on each $s'$ and Assumption 1, and the last inequality holds by Cauchy-Schwarz inequality. $\qquad\square$

## E.4 Proof of Lemma 3

*Proof.* We have that

$$\sum_{h \in [H]} \sum_{s,a} \hat{\xi}_h^{\hat{\pi}^*}(s,a) \mathrm{Var}_{P(s,a)}\left( V_{h+1}(s')^{2^i} \right)$$

$$= \sum_{h \in [H]} \sum_{s,a} \hat{\xi}_h^{\hat{\pi}^*}(s,a) \left[ \sum_{s'} P(s'|s,a) V_{h+1}(s')^{2^{i+1}} - \left( \sum_{s'} P(s'|s,a) V_{h+1}(s')^{2^i} \right)^2 \right]$$

$$= \sum_{h \in [H]} \sum_{s,a} \hat{\xi}_h^{\hat{\pi}^*}(s,a) \left[ \sum_{s'} \Big( P(s'|s,a) - \hat{P}(s'|s,a) \Big) V_{h+1}(s')^{2^{i+1}} + V_h(s)^{2^{i+1}} - \left( \sum_{s'} P(s'|s,a) V_{h+1}(s')^{2^i} \right)^2 \right]$$

$$- \sum_{s} \mu(s) V_1(s)^{2^{i+1}}$$

$$\leq \sum_{h \in [H]} \sum_{s,a} \hat{\xi}_h^{\hat{\pi}^*}(s,a) \left[ \sum_{s'} \Big( P(s'|s,a) - \hat{P}(s'|s,a) \Big) V_{h+1}(s')^{2^{i+1}} \right]$$

$$+ \sum_{h \in [H]} \sum_{s,a} \hat{\xi}_h^{\hat{\pi}^*}(s,a) \left[ V_h(s)^{2^{i+1}} - \left( \sum_{s'} P(s'|s,a) V_{h+1}(s') \right)^{2^{i+1}} \right]$$

$$\leq \sum_{h \in [H]} \sum_{s,a} \hat{\xi}_h^{\hat{\pi}^*}(s,a) \left[ \sum_{s'} \Big( P(s'|s,a) - \hat{P}(s'|s,a) \Big) V_{h+1}(s')^{2^{i+1}} \right]$$

$$+ 2^{i+1} \sum_{h \in [H]} \sum_{s,a} \hat{\xi}_h^{\hat{\pi}^*}(s,a) \left[ V_h(s) - \left( \sum_{s'} P(s'|s,a) V_{h+1}(s') \right) \right],$$

where in the last step, we use the fact that $\hat{\pi}^*$ is a deterministic policy, and

$$V_h^{\pi^*}(s) \geq r(s,a) + \sum_{s'} P(s'|s,a) V_{h+1}^{\pi^*}(s'), \quad \forall a \in \mathcal{A},$$

$$V_h^{\hat{\pi}^*}(s) = r(s, \hat{\pi}^*(s)) + \sum_{s'} P(s'|s, \hat{\pi}^*(s)) V_{h+1}^{\pi^*}(s'),$$

which guarantees that for $V_h(s) = V_h^{\pi^*}(s)$ and $V_h(s) = V_h^{\pi^*}(s) - V_h^{\hat{\pi}^*}(s)$,

$$\hat{\xi}_h^{\hat{\pi}^*}(s,a) \left[ V_h(s) - \left( \sum_{s'} P(s'|s,a) V_{h+1}(s') \right) \right] \geq 0.$$

Moreover, we have that

$$\sum_{h\in[H]}\sum_{s,a}\hat{\xi}_h^\pi(s,a)\left[V_h(s)-\sum_{s'}\hat{P}(s'|s,a)V_{h+1}(s')\right]$$

$$=\sum_{h\in[H]}\sum_{s}\hat{\xi}_h^\pi(s)V_h(s)-\sum_{h\in[H]\setminus\{1\}}\sum_{s}\hat{\xi}_h^\pi(s)V_h(s)$$

$$=\sum_{s}\mu(s)V_1(s).$$

So we can conclude that

$$\sum_{h\in[H]}\sum_{s,a}\hat{\xi}_h^\pi(s,a)\left[V_h(s)-\left(\sum_{s'}P(s'|s,a)V_{h+1}(s')\right)\right]$$

$$=\sum_{h\in[H]}\sum_{s,a}\hat{\xi}_h^\pi(s,a)\sum_{s'}\left[\hat{P}(s'|s,a)-P(s'|s,a)\right]V_{h+1}(s')+\sum_{s}\mu(s)V_1(s),$$

thus finish the proof. $\qquad\square$

### E.5 Proof of Lemma 4

*Proof.* Lemma 4 have been shown in Dann et al. [51, Lemma E.15]. Here we include the proof for completeness.

$$v^\pi-\hat{v}^\pi$$

$$=\sum_{s,a}\hat{\xi}_1^\pi(s,a)(Q_1^\pi(s,a)-\hat{Q}_1^\pi(s,a))$$

$$=\sum_{s,a}\left[\hat{\xi}_1^\pi(s,a)\left(r(s,a)-\hat{r}(s,a)+\sum_{s'}[P(s'|s,a)V_2^\pi(s')]-\sum_{s'}\left[\hat{P}(s'|s,a)\hat{V}_2^\pi(s'))\right]\right)\right]$$

$$=\sum_{s,a}\left[\hat{\xi}_1^\pi(s,a)\left(r(s,a)-\hat{r}(s,a)+\sum_{s'}\left[(P(s'|s,a)-\hat{P}(s'|s,a))V_2^\pi(s')\right]\right)\right]+\sum_{s}\hat{\xi}_2^\pi(s)(V_2^\pi(s)-\hat{V}_2^\pi(s))$$

$$=\cdots$$

$$=\sum_{h\in[H]}\sum_{s,a}\left[\hat{\xi}_h^\pi(s,a)\left(r(s,a)-\hat{r}(s,a)+\sum_{s'}\left[(P(s'|s,a)-\hat{P}(s'|s,a))V_{h+1}^\pi(s')\right]\right)\right]$$

$\qquad\square$

### E.6 Proof of Lemma 5

*Proof.* By Bernstein's inequality, we know that with high probability,

$$\left|\sum_{h\in[H]}\sum_{s,a}\hat{\xi}_h^\pi(s,a)\left[r(s,a)-\hat{r}(s,a)\right]\right|$$

$$\leq\sum_{h\in[H]}\sum_{s,a}\hat{\xi}_h^\pi(s,a)\left[\sqrt{\frac{r(s,a)\iota}{n(s,a)}}+\frac{\iota}{n(s,a)}\right]$$

$$\leq\sqrt{\frac{\iota}{Kd_m}}\sqrt{\sum_{h\in[H]}\sum_{s,a}\hat{\xi}_h^\pi(s,a)r(s,a)}+\frac{\iota}{Kd_m},$$

where the first inequality use the fact that $r(s,a)\leq 1$ and the second inequality is Cauchy-Schwarz inequality associated with Assumption 2. With Assumption 1, we know that

$$\sum_{h\in[H]}\sum_{s,a}\hat{\xi}_h^\pi(s,a)r(s,a)\leq 1,$$

as each trajectory that can generated by $\widehat{\mathcal{M}}$ can be generated by $\mathcal{M}$, which finishes the proof. $\quad\square$

### E.7 Proof of Lemma 6

*Proof.* Lemma 6 is similar to the Lemma 11 in Zhang et al. [27], and here we provide a simplified proof. Define

$$i_0 := \max\{i \mid 2^{2i}\lambda_2^2 \le \lambda_1 \mathbb{V}(i)\} \cup \{0\},$$

which is the largest integer satisfies the inequality $2^{2i}\lambda_2^2 \le \lambda_1 \mathbb{V}_1(i)$. We will make the recursion at most $i_0$ times. As $\mathbb{V}(i)$ is upper bounded by $H$, we know $i_0 = O\left(\log \frac{H\lambda_1}{\lambda_2^2}\right)$. If $i_0 = 0$, we have that

$$\mathbb{V}(1) \le \frac{4\lambda_2^2}{\lambda_1}.$$

Otherwise,

$$
\begin{aligned}
&2^{2i_0}\lambda_2^2\\
\le&\lambda_1 \mathbb{V}(i_0)\\
\le&\lambda_1\left(\sqrt{\lambda_1 \mathbb{V}(i_0+1)} + \lambda_1 + 2^{i_0+1}\lambda_2\right)\\
\le&\lambda_1\left(\sqrt{2^{2i_0+2}\lambda_2^2} + \lambda_1 + 2^{i_0+1}\lambda_2\right)\\
=&\lambda_1(2^{i_0+2}\lambda_2 + \lambda_1),
\end{aligned}
$$

we have that

$$(2^{i_0}\lambda_2 - 2\lambda_1)^2 \le 5\lambda_1^2,$$

which means

$$2^{i_0}\lambda_2 \le (\sqrt{5}+2)\lambda_1,$$

so $\forall i \le i_0$, we have that

$$\mathbb{V}(i) \le \sqrt{\lambda_1 \mathbb{V}(i+1)} + (\sqrt{5}+3)\lambda_1.$$

As

$$\mathbb{V}(i_0) \le 2^{i_0+2}\lambda_2 + \lambda_1 \le (4\sqrt{5}+9)\lambda_1,$$

and when $\mathbb{V}(i+1) \ge \left(\frac{1}{2} + \sqrt{\sqrt{5}+\frac{13}{4}}\right)\lambda_1$, we have $\mathbb{V}(i) \le \mathbb{V}(i+1)$, when $\mathbb{V}(i+1) \le \left(\frac{1}{2} + \sqrt{\sqrt{5}+\frac{13}{4}}\right)\lambda_1$, $\mathbb{V}(i) \le \left(\frac{1}{2} + \sqrt{\sqrt{5}+\frac{13}{4}}\right)\lambda_1$. So $\mathbb{V}(1) \le (4\sqrt{5}+9)\lambda_1$. Combine the two cases, we have that

$$\mathbb{V}(1) \le \max\left\{\frac{4\lambda_2^2}{\lambda_1}, (4\sqrt{5}+9)\lambda_1\right\},$$

which means

$$\mathbb{V}(0) \le \max\left\{4\lambda_2 + \lambda_1, \left(\sqrt{4\sqrt{5}+9}+1\right)\lambda_1 + 2\lambda_2\right\} \le 6(\lambda_1+\lambda_2),$$

which concludes the proof. □

## F  Further Discussion

### F.1  Time-Homogenous vs. Time-Inhomogenous

#### F.1.1  Offline Policy Evaluation

One can notice that the lower bound of offline policy evaluation under time-homogenous setting (shown above) and time-inhomogenous setting (that can be simplified from the Cramer-Rao lower

bound of [13]) are identical. It can be surprising at the first glance, as one common belief for reinforcement learning is that the time-homogenous MDP should be easier than the time-inhomogenous MDP. However, we remark that, this argument does not hold for the policy evaluation. This is due to the fact that in time-homogenous setting, as the error from the estimation of transition will be accumulated along the horizon, we need $\Omega(H)$ samples for each transition to make sure that the accumulated error should be $O(1)$. However, in the time-inhomogenous setting, the error from the estimation of each level transition is probably not accumulated, which means we only need $\Omega(1)$ samples for each transition at each level to make sure the final error to be $O(1)$. This can also be seen in the analysis of offline policy evaluation under time-inhomogenous setting in [19, 24], where the authors decouple the error from each sample $(s_h, a_h, s_{h+1})$, which forms a martingale difference sequence with each step variance $O(1)$ that can then apply the Freedman's inequality to obtain a tight bound. Under time-inhomogenous setting, we notice that the error from each sample $(s, a, s')$ can only form a martingale difference sequence with each step variance $O(H)$, which will not lead to a tighter bound. We want to emphasize that *the results in [19, 24] does not directly indicate results in this paper*.

Moreover, we notice that the analysis in [19, 24] can be translated to a value-dependent bound and thus can be horizon-free under Assumption 1, with the following lemma:

**Lemma 12.** *Under Assumption 1, we have that*

$$\sum_{h \in [H]} \sum_{s,a} \xi_h^\pi(s,a) Var(r(s,a) + V_{h+1}^\pi(s')) \le 3v^\pi,$$

*where the term at the left hand side is exactly the variance term of MIS estimator considered in [19, Lemma 3.4]*

*Proof.*

$$\sum_{h \in [H]} \sum_{s,a} \xi_h^\pi(s,a) Var\left(r(s,a) + V_{h+1}^\pi(s')\right)$$

$$\le \sum_{h \in [H]} \sum_{s,a} \xi_h^\pi(s,a) \left[ r(s,a) + \sum_{s'} P(s'|s,a)[V_{h+1}^\pi(s')]^2 - \left(\sum_{s'} P(s'|s,a)V_{h+1}^\pi(s')\right)^2 \right]$$

$$\le \sum_{h \in [H]} \sum_{s,a} \xi_h^\pi(s,a) \left[ r(s,a) + Q_h^\pi(s,a)^2 - \left[\sum_{s'} P(s'|s,a)V_{h+1}^\pi(s')\right]^2 \right]$$

$$\le 3 \sum_{h \in [H]} \sum_{s,a} \xi_h^\pi(s,a)r(s,a) = 3v^\pi,$$

where we use the following fact:

$$\sum_{s,a,s'} \xi_h^\pi(s,a)P(s'|s,a) \left[V_{h+1}^\pi(s')\right]^2$$

$$= \sum_{s'} \xi_{h+1}^\pi(s') \left[V_{h+1}^\pi(s')\right]^2$$

$$= \sum_{s'} \xi_{h+1}^\pi(s') \left[\sum_{a'} \pi(a'|s')Q_{h+1}^\pi(s',a')\right]^2$$

$$\le \sum_{s'} \xi_{h+1}^\pi(s') \sum_{a'} \pi_{h+1}(a'|s')Q_{h+1}^\pi(s',a')^2$$

$$= \sum_{s',a'} \xi_{h+1}^\pi(s',a')Q_{h+1}^\pi(s',a')^2.$$

$\square$

This shows that the offline policy evaluation under time-inhomogeneous setting is also horizon-free, which also matches the intuition that if the density ratio can be lower bounded, then we can construct

a trajectory-wise importance sampling (IS) estimator that only depends on the number of episodes. Notice that, in [19, 24], the higher order term has an additional $\sqrt{SA}$ factor. Tightening such factor is beyond the scope of this paper.

### F.1.2 Offline Policy Optimization

For the offline policy optimization, it's known that time-inhomogeneous MDP cannot avoid the additional $H$ factor in sample complexity, thus can never be horizon-free. We also want to remark that with Lemma 12, the results in [24] can be translated to a $\tilde{O}\left(\sqrt{\frac{H^2}{nd_m}}\right)$ error bound as well as a $\tilde{O}\left(\frac{H^2}{d_m\epsilon^2}\right)$ sample complexity bound that holds for all range of $\epsilon \in (0, 1]$. Our bound, however, have an additional higher order term $\tilde{O}\left(\frac{SH}{d_m\epsilon}\right)$. We would like to remark that, this is due to the time-homogeneous nature of our setting, which makes $V_h^{\hat{\pi}^*}$ heavily depends on $\hat{P}$. On the other hand, in time-inhomogeneous setting, $V_h^{\hat{\pi}^*}$ only depends on $\hat{P}_k(s, a)$ for $k > h$, thus can directly apply Bernstein's inequality when bounding the term $\left(\hat{P}_h(s, a) - P_h(s, a)\right)V_{h+1}^{\hat{\pi}^*}$, which will not introduce additional $S$ factor. The best known result for finite horizon time-homogenous MDP [27] also has this additional $S$ factor, and how to eliminate this additional $S$ factor remains an open problem.

## F.2 Finite-Horizon vs. Infinite-Horizon

### F.2.1 Offline Policy Evaluation

We remark that [28] provides a value-dependent bound for the policy evaluation under infinite-horizon generative model setting that accommodates full range of $\epsilon$. We obtain the similar results in finite-horizon setting that can accommodate full range of $\epsilon$, however, with a different and probably simpler analysis.

### F.2.2 Offline Policy Optimization

[28] also provides a minimax-optimal sample complexity bound up to logarithmoc factors for policy optimization with generative model that accommodates full range of $\epsilon$. We notice that such kinds of analysis cannot be directly applied to the finite-horizon setting, as their "absorbing MDP" technique cannot be directly applied to the finite-horizon MDP, due to the difference of time-homogenous value function in infinite-horizon setting and time-inhomogenous value function in finite-horizon setting, which has been pointed out by [1]. And thus most of the existing work does not provide a sample complexity bound that can match the lower bound. To the best of our knowledge, our work first provide a sample complexity bound that match the lower bound up to logarithmic factors and an high-order term.

## F.3 With General Function Approximation

There are also works considering the offline policy optimization with general function approximation under different kinds of function class assumption like realizability and completeness [e.g. 52–54], which generally do not imply tight bounds under certain scenarios like e.g. tabular MDP. We leave the extension to general function approximation as future work.

## F.4 Without Sufficient Exploratory Data

Recently, [45, 46] also introduces another perspective on performing offline policy optimization within a local policy set when the offline data is not sufficient exploratory, which is different from the global policy optimization we consider here. We want to emphasize that, if we want to approach the global optimal policy, our assumption on good data coverage *i.e.* Assumption 2 is necessary. Otherwise, we will suffer from the error from under-explored state-action pair, as Theorem 2 and Theorem 4 suggests.