# OpenReview forum: "Nearly Horizon-Free Offline Reinforcement Learning"
_NeurIPS.cc/2021/Conference — NeurIPS 2021 Poster_

### Official Review · Reviewer_ND3y · 2021-07-14

**Rating:** 6
**Confidence:** 3

**Summary:**

This paper provides near-matching, horizon-free upper and lower-bounds on the Offline Policy Evaluation (OPE) and Offline Policy Optimization (OPO), in finite horizon time-homogeneous MDPs, with tabular and linear MDP structures with anchor points.

**Main Review:**

The presentation of the paper is clear and illustrative, therefore I believe substantial improvements have been made since the previous submission in terms of paper polishing.

The additional setting of linear MDP with anchor points, on the other hand, seems not substantial. It's not a well-accepted setting and the analysis seems to be a straightforward extension of the tabular analysis, since after all anchor point means only d (s,a) pairs matter and so everything can be represented back into a table. Unlike the ICML reviewers, I think even the tabular results alone are worth publishing as a theoretical paper.

Besides that my main complaint is also on the strong assumption of uniform data coverage, which is too strong for most applications. A recent line of work performs offline reinforcement learning without a uniformly lower-bounded data distribution with the principle of pessimism. It has been shown that the optimal policy can be found as long as the offline data properly covers the visitation distribution of the optimal policy. Can the author comment on whether horizon-free learning possible without the uniform data coverage assumption? Also, this line of work is also worth covering in the related work section.

**Time Spent Reviewing:**

1

---

> ### Author Response · Authors · 2021-08-09
> **Thanks for your kind review. We discuss the relation between different kinds of assumptions below.**
>
> We thanks for the kind comments and suggestions on our paper. We find [1] and [2] define the data coverage via covering the visitation of optimal policy. This kind of definition is indeed similar to our assumption. Notice that if the behavior policy covers the state-action that can be visited by the optimal policy, we can consider a new MDP that only consists the state-action that can be visited by the behavior policy. If we can find the optimal policy in this new MDP, we will know that this policy is optimal on the original MDP. This is due to the fact that even we can visit some states not exist in the new MDP, it will not lead to a better performance as we know the optimal policy will not visit these states. In this sense, it's possible to identify the optimal policy if the behavior policy only covers the optimal policy. It's also possible to directly translate our current analysis to this new setting with $d_m$ replaced by the minimum visiting probability on this new MDP. We will cover this lines of work in the related work in the next revision.
>
> [1] Liu, Yao, Adith Swaminathan, Alekh Agarwal, and Emma Brunskill. Off-policy policy gradient with state distribution correction. arXiv preprint arXiv:1904.08473 (2019).
>
> [2] Yin, Ming, Yu Bai, and Yu-Xiang Wang. Near-optimal offline reinforcement learning via double variance reduction. arXiv preprint arXiv:2102.01748 (2021).

---

### Official Review · Reviewer_PYbq · 2021-07-15

**Rating:** 9
**Confidence:** 3

**Summary:**

The authors revisit offline RL on episodic MDPs both in the tabular and linear with anchors cases. They prove complexity bounds that are nearly horizon $H$ free (up to polylog terms in the horizon $H$), when the maximum return is 1 for both offline policy evaluation and offline policy optimization.

**Limitations And Societal Impact:**

yes

**Main Review:**

This is a remarkable theoretical paper. I took a great interest reviewing it and I think it will have a great impact in the offline policy evaluation/optimization community.

Major remarks:
- The writing around the assumptions is not as sharp as everywhere else:
   - line 197: almost surely -> It is with respect to the stochasticity or R and P? But since they are defined on a finite MDP, and since it is a finite sum, either this event is $0$ or $>0$.
   - lines 201-202: Assumption 1 is equivalent to the uniformly bounded reward isn't it? When moving from one to another, and back, you lose a factor $H$, but otherwise, they are equivalent.
   - line 203: it can be generalized but as I said in my previous remark, a factor $H$ appears, so it defeats the main arguments of the method.
   - Assumption 2 seems unnecessary to me. The actual assumption that matters and that is not formalized as such (even though it is well recalled in theorems), is that $K= \tilde{\Omega}(1/d_m)$.
- the actual dependency in $H$ is actually hidden in $d_m$: the longest horizon the MDP is the more skewed is going to be the state visitation distribution of any policy. And to prove that, many papers have shown that model-based planning (which may be wrongly called, because performing q-learning on the samples themselves until full convergence would lead to the same result) lead to worse performance than pessimistic/conservative planning **on average**. Maybe the authors should be more explicit about this effect.

Minor remarks:
- lines 64-66: I could not understand this sentence.
- line 98: refers to assumption 1, while it is only enounced 2 pages later.
- table 1: you use $\epsilon$ in Table 1, but it is introduced only line 113. Also, the writing about table 1 does not use the same parameters for the bounds, so it is a bit inconsistent.
- line 155: even -> even though.
- line 158: of $\pi$ -> delete
- lines 171-172: $(s_i=s,a_i=a,r_i,s'_i)$ and $(s_i=s,a_i=a,r_i,s'_i=s')$
- lines 275-276: local policy optimization was already done by Kakade2002, Petrik2016, and Laroche2019, just to name a few.
- lines 282-284 are a repetition of lines 280-281?


[Kakade2002] Kakade, S., & Langford, J. (2002). Approximately Optimal Approximate Reinforcement Learning. In International Conference on Machine Learning.

[Petrik2016] Petrik, M., Ghavamzadeh, M., & Chow, Y. (2016). Safe policy improvement by minimizing robust baseline regret. In Advances in Neural Information Processing Systems (pp. 2298-2306).

[Laroche2019] Laroche, R., Trichelair, P., & Des Combes, R. T. (2019, May). Safe policy improvement with baseline bootstrapping. In International Conference on Machine Learning (pp. 3652-3661).

**Time Spent Reviewing:**

3

---

> ### Author Response · Authors · 2021-08-09
> **Thanks for your detailed and positive review. We address the major concerns in the following.**
>
> We thank for the detailed comments and suggestions on our paper. We address the major remarks in the following:
> * Questions on the assumptions:
>     + Line 197: Your understanding is correct, but we would like to kindly argue that \emph{almost surely} here is still formal and will not have any unnecessary ambiguity for the potential reader.
>     + Line 201-203: It in fact is not equivalent to the uniformly bounded reward, as our Assumption 1 allows for the sparse reward while the uniformly bounded reward assumption does not allow. We can consider a simple MDP that only have one action and deterministic transition. If we can only obtain a reward of $1$ at the final step, it indeed satisfies Assumption $1$ but not the uniformly bounded reward. So our Assumption is more general than the uniformly bounded reward, and our analysis cannot use the structure of the uniformly bounded reward, which is technically more challenging.
>     + Assumption 2: We tend to have the Assumption 2 here, as we generally need a data coverage assumption under the offline setting. We will highlight the condition $K = \tilde{\Omega}(\frac{1}{d_m})$ in the next revision.
> * The issue on $d_m$: We want to kindly remark that, $d_m$ is generally not a term we can control, and thus we treat it as an universal constant and consider what we can do with a fixed $d_m$. As we have discussed in Line 208-213, the quality of behavior policy determines the value of $d_m$. If we want to change $d_m$, we are in fact deal with an online problem, which is out of the scope of this paper. We also want to remark that, if the behavior policy is sufficient exploratory, then $d_m$ is not necessarily scale with $H$.
> * On the pessimistic/conservative planning: We notice that there are several empirical works discussing the importance of pessimism in offline setting (e.g. [1] and [2]), and it's possible that pessimistic/conservative planning can achieve better empirical performance than the plug-in approach when the data coverage is not good, as it can avoid the issue introduced by the poor estimation on the unseen state-action pairs. We will discuss these work in our next revision. However, we want to kindly remark that both pessimism planning and plug-in approach are minimax-optimal under our setting, and we believe finding the exact conditions in which one method provably outperforms the other one is out of the scope of our paper.
>
> We also thank for pointing out the minor issues and related work. And we want to clarify that Line 282-284 are different from Line 280-281 as Line 282-284 are talk about term (5) and (6) individually which are introduced after Line 280-281. We will revise our paper accordingly.
>
> [1] Buckman, Jacob, Carles Gelada, and Marc G. Bellemare. "The Importance of Pessimism in Fixed-Dataset Policy Optimization." International Conference on Learning Representations. 2020.
>
> [2] Kumar, Aviral, et al. "Conservative q-learning for offline reinforcement learning." arXiv preprint arXiv:2006.04779 (2020).

---

> > ### Comment · Reviewer_PYbq · 2021-08-11
> > **discussion 1**
> >
> > - line 197: I agree it is sound, but unnecessarily confusing. I would still advise the authors to remove a.s.
> > - line 201-203: I don't get your example: the reward is still uniformly bounded by 1 (not $1/H$). As I said in my first comment, there is a scaling factor $H$ appearing, but otherwise, they are equivalent in my understanding.
> > - $H$ *hidden* in $d_m$: I agree with what you say. My point is that there will be in general an empirical correlation between $H$ and $d_m$ (which is hard to be quantified). So maybe *hidden* is too strong of a word.

---

> > > ### Author Response · Authors · 2021-08-11
> > > **Thanks for your quick response. We address your concern below.**
> > >
> > > * Line 197: We agree that there can be some potential misunderstanding here, and we will give more discussion on the Assumption 1 in the next revision.
> > > * Line 201-203:
> > >     + Here we refer the uniformly bounded reward as per-step reward upper bounded by $\frac{1}{H}$, so that both Assumption 1 and uniformly bounded reward have a bounded total reward of $1$, which makes the comparison fair. We will revise our paper to make this clear.
> > >     + We guess you are wondering if Assumption 1 implies uniformly bounded reward. In fact, if the total reward is bounded by $1$, then per-step reward should also be bounded by $1$ as we assume the reward is non-negative. However, without Assumption $1$, uniformly bounded reward can have total reward as large as $H$, but in our Assumption, the total reward is upper bounded by $1$. In this case, they cannot translate to each other equivalently.
> > >     + In summary, our Assumption 1 assumes per-step reward is upper bounded by $1$, and total reward is upper bounded by $1$. Uniformly bounded reward assumes per-step reward is upper bounded by $\frac{1}{H}$, which implies total reward is upper bounded by $1$. These two assumptions are not equivalent and cannot be transformed with a single scaling.
> > > * $H$ and $d_m$: We agree that *hidden* is a too strong word. As we have said, it's a term that we cannot control, so we only treat it as a constant, without further considering it's dependency on the other terms.
> > >
> > > Thanks for your quick response. We hope our response can address your major concern.

---

> > > > ### Comment · Reviewer_PYbq · 2021-08-11
> > > > **discussion 2**
> > > >
> > > > - lines 201-203: there is an equivalence between *bounded reward* and *bounded total reward*, but with a loss of scale $H$, meaning that applying your *bounded total reward* theorem to a *bounded reward* setting either restricts too much the set of admissible problems by only considering rewards that are bounded by $1/H$, or incurs the apparition of a factor $H$ in the bounds by considering rewards that are bounded by $1$, which defeats the bound complexity improvement claimed in the paper. This is why I think that the sentence lines 202-203 is misleading. I do not consider this as a major issue since most relevant problems are task based (rewards 0 everywhere except at task completion where it's 1).
> > > > - $H$ and $d_m$: *hidden* was an unfortunate choice of words, sorry for that. My point was to advocate for some sentence(s) that would explain why every offline RL practitioner empirically observed a significant dependency on $H$.
> > > > - (digression) $d_m$ is a term that cannot be controlled: I disagree with this. [Laroche2019] showed that it could be controlled in a safe policy improvement setting by only allowing policy changes where $n(s,a)$ is larger to some hyperparameter $n_m$. This is equivalent to choosing $d_m = n_m/n$ as a hyperparameter. Obviously the global near-optimality cannot be controlled anymore, but the safe policy improvement over the behavioral policy can, as well as the near optimality under the constrained policy search. This is cool actually.
> > > >
> > > > These are not major concerns, I don't have any. Congratulations for your paper that I would be happy to cite in the future.

---

> > > > > ### Author Response · Authors · 2021-08-11
> > > > > **Thanks for your comments. We illustrate our idea below.**
> > > > >
> > > > > * Line 201-203: Sorry for the confusion we made in Line 202-203. Indeed we want to argue that our analysis can be generalized to any uniformly bounded reward (e.g. reward upper bounded by $1$) by properly normalizing the total reward to satisfy bounded total reward and applying our analysis with an additional factor on the total reward. We don't need to necessarily constrain the per-step reward upper bounded by $\frac{1}{H}$. We still want to kindly remark that our bounded total reward assumption is totally different from the bounded per-step reward, as we show that it's possible to have the instances that total reward is bounded by $1$ but per-step reward is not upper bounded by $\frac{1}{H}$. And it's probable to generalize our methods to bounded per-step reward setting, but not vice-versa. We will make this more clear in the next revision.
> > > > >
> > > > > * $H$ and $d_m$: We will revise our paper based on your suggestion. Indeed it's quite important to make the practitioner understand our work properly.
> > > > >
> > > > > * Controlled $d_m$: Thanks for pointing this interesting paper out. Other reviewers (e.g. Reviewer ND3y) also point out some different definitions of $d_m$. As we focused on identifying global policies, we need to define $d_m$ in such way. However, if we only want to improve the performance or identify the optimal policy in a sub-MDP, then we can have a different definition of $d_m$ and a different set of theorems. We will provide more discussions on this in the next revision.
> > > > >
> > > > > We want to thank for your positive review and suggestions on improving our paper again. We will carefully revise our paper accordingly.

---

### Official Review · Reviewer_aVzM · 2021-07-16

**Rating:** 6
**Confidence:** 3

**Summary:**

This paper provided horizon-free theoretical results for offline policy evaluation and improvement. It showed the bound for estimation error and sub-optimality under several assumption.

**Limitations And Societal Impact:**

The author addressed their limitation adequately.

**Main Review:**

Originality. The proof of their results is origin.

Quality. This is a complete work. The showed both upper bound and lower bound under their assumption. However, the assumption might be too strong. Apart from the assumption previously notice, time-homogeneous is also strong since some other work on offline RL does need it.

Clarity. This paper is well-written.

Significance. This paper is significance in the sense that horizon-free results can deepen our understanding for episodic RL.

**Time Spent Reviewing:**

6

---

> ### Author Response · Authors · 2021-08-09
> **Thanks for your review. We discuss the generality of the assumption we use below.**
>
> We thank for your time spent on our paper. For the assumptions, we would like to kindly argue that our assumptions are fairly standard in the related literature. For example, Assumption 1 and 3 has been used in several previous work, e.g. Assumption 1 appears in [1] and [2], Assumption 3 has been considered in [3] and [4]. Meanwhile our lower bound shows Assumption 2 is necessary under the minimax sense and similar condition also appears in the previous work like [5]. Time-homogeneity is also a common assumption that also considered in [1] and [2], and in practice many reinforcement learning problems like games are time-homogeneous.
>
> We are improving the sample complexity to the theoretical lower bound under standard assumptions in the literature for clearly comparison. We hope our clarification can address your concerns, so that you may adjust your score.
>
> [1] Ruosong Wang, Simon S Du, Lin F Yang, and Sham M Kakade. Is long horizon reinforce- ment learning more difficult than short horizon reinforcement learning? arXiv preprint arXiv:2005.00527, 2020.
>
> [2] Zihan Zhang, Xiangyang Ji, and Simon S Du. Is reinforcement learning more difficult than ban- dits? a near-optimal algorithm escaping the curse of horizon. arXiv preprint arXiv:2009.13503, 2020.
>
> [3] Lin Yang and Mengdi Wang. Sample-optimal parametric q-learning using linearly additive features. In International Conference on Machine Learning, pages 6995–7004. PMLR, 2019.
>
> [4] Qiwen Cui and Lin F Yang. Is plug-in solver sample-efficient for feature-based reinforcement learning? arXiv preprint arXiv:2010.05673, 2020.
>
> [5] Ming Yin and Yu-Xiang Wang. Asymptotically efficient off-policy evaluation for tabular reinforcement learning. In Silvia Chiappa and Roberto Calandra, editors, Proceedings of the Twenty Third International Conference on Artificial Intelligence and Statistics, volume 108 of Proceedings of Machine Learning Research, pages 3948–3958. PMLR, 26–28 Aug 2020

---

> > ### Comment · Reviewer_aVzM · 2021-08-19
> > **Thanks for your response**
> >
> > Thanks for your response! I think the response address my concern and will raise my score accordingly.

---

> > > ### Author Response · Authors · 2021-08-19
> > > **Thanks for your reply!**
> > >
> > > We would like to thank you for your reply, as well as your adjustment on the score based on our response.

---

### Official Review · Reviewer_vD82 · 2021-08-03

**Rating:** 6
**Confidence:** 3

**Summary:**

The paper analyzes the plug-in approach to off-policy evaluation and optimization in reinforcement learning, and sharpens some of the sample complexity bounds (assuming sufficiently exploratory data, time-homogenous MDPs). The key novelty is in upper-bounding the variance of estimated values cleverly by using higher-order powers of appropriate values. By doing so, the paper's derived bounds do not directly depend on the horizon of the MDP. One benefit of the proposed analysis is that a very similar line of reasoning is sufficient to derive horizon-independent bounds for both evaluation and optimization, as well as for tabular MDPs as well as linear MDPs with additional structure.

**Ethical Concerns:**

The paper does not raise any ethical concerns at this time, nor does it increase the scope of RL applications that can be attempted in the future (for instance, no new applications that may be problematic).

**Limitations And Societal Impact:**

The authors do not address the societal impact of their contributions. However, this is a completely theoretical work that aims to sharpen the analysis of an existing algorithm, incremental improvement by using a known technique. So there societal impact of this work is unlikely to be adverse.
Re: the limitations, the authors do not discuss the plausibility of their assumptions nor the motivations of the linear MDP w/ anchor points setting.

**Main Review:**

(After author feedback)
The authors' responses adequately addressed questions around significance, motivation and technical concerns. There were remaining questions about the novelty/originality and scope of contributions. The paper can be substantially stronger by addressing them (e.g. by weakening Assumption 2).

-----
The paper had a reasonable exposition, and the high-level ideas were explained clearly. This review focuses only on the weaknesses.

Correctness: Stepping through the appendix, there were two confusing points (detailed below). This makes me confused about Theorem 1 and 2. If this confusion is clarified, the paper would improve to be borderline. Given the confusions, I have not carefully checked the proofs of the policy optimization theorems.

Originality: The main idea is to use the total variance bounding technique of [27] in the off-policy setting. This is done in a pretty brute-force way (assume that there is sufficient exploration via Assumption 2). The derivations become somewhat more tedious (keeping track of the coverage coefficient in each step) but not novel.

Significance: Disappointed to see that motivation of the work was not clear. Beyond an academic exercise in sharpening the upper bounds, does this help explain some previously unexplained phenomenon? For instance, in practical applications of the plug-in approach, do we see that it works surprisingly well as H increases (while existing theory suggested that it should not)?
Are there practical applications where Assumptions 1 and 2 (and for linear MDPs, the anchor data assumption) can reliably hold? Under the exploratory data assumption, would other approaches have a H-dependence even in time-homogenous MDPs? For instance, an off-policy policy gradient approach might depend on H while the plug-in approach does not (and this might guide us to choose the latter in large-H regimes).
The anchor data assumption and construction was unusual to me. Are there practical applications where they are useful? (If not, they seemed to be just assumptions of convenience so as to transport results from tabular MDPs to linear MDPs without having to construct an epsilon-net).
The paper unfortunately does not address any of these questions to establish the motivation or significance of its contributions.

Confusions:
Appendix B [Proof of Theorem 2]
0 <= c_1 <= 1. And 0 <= c_2 <= c_1. In Line 619, when setting c_2 = c_1 - ...(H), note that c_2 < 0 for large values of H which is not allowed.
But c_2 has to be set to such a term to witness a value gap that scales with H which is needed to assert the claim in Line 622.
Please clarify? I am skeptical that this MDP construction will witness an Omega(sqrt(H/(n*d_m))) estimation gap under Assumption 1 and 2.

Appendix A [Line 555]
Perhaps derive V_1(i) <= H for all i using Assumption 1 to make this more explicit.
This statement is easy to show under the uniform reward assumption that r_h < 1/H.
However under assumption 1, it takes more to assert this claim constructively. For example, time-homogeneity rules out situations where r_H = 1 and r_h =0 for h < H; and situations where a set of states only become reachable at H. In such situations however, this claim is false. Does time-homogeneity rule out all pathological situations where V_1(i) > H for some i?

Minor issues:
- Since Table 2 and Section 2 require knowing what Assumption 1 refers to, it may be better to define it in Sec2, and simply re-state it in Sec 4.2.
- Notation collision: r is defined as E[R] above. Unfortunately, r_i \sim P(R). This makes some objects ambiguous.
For instance, the empirical reward in Line 186 should be hat{r}(s,a) = \sum_i r_i 1{...}/n(s,a). As written currently, r(s_i, a_i) can be misunderstood as the expected reward of (s_i, a_i) rather than the realized reward r_i. This also affects the statement of Assumption 1 (what is r_h exactly? is there a typo in Line 198, r_i should be r_h?).
- Clarify the empirical MDP construction hat{M} for P(.|s,a) and R(s,a) when n(s,a)=0. Or specify the regime (under Assumption 2) clearly.
Perhaps bring Assumption 2 into Sec 3.2?

Typos:
Line 45 - related *line* of work
Line 47 - only *has* poly log
Line 65 - that *has* broadly
Line 84 - we *need* to deal
Line 155 - *even though P and R remain invariant when h changes*
Line 158 - cumulative *reward under policy \pi* is defined
Line 165 - perspective *on* policy evaluation
Line 177 - *computationally* efficient
Line 212 - min_s,a n(s,a) \approx H*K*d_m. The \forall s,a is superfluous.
Footnote 3 - lower bound *has* been provided

**Time Spent Reviewing:**

7

---

> ### Author Response · Authors · 2021-08-09
> **Thanks for your detailed and constructive review. We first address the technical concern in the following. (1/2)**
>
> We thank for the detailed comments and suggestions on our paper.
>
> * Regarding the originality:
>     We want to first re-emphasize that our method is not brute-force nor tedious. In fact, Assumption 2 is a common assumption in offline reinforcement learning, which also appears in e.g. [1], that is not only "brutally" introduced to suit our technical analysis. Our minimax lower bound in Theorem 2 and Theorem 4 also indicates the importance of such coverage coefficient. Moreover, as we have argued in Line 78-85, our method is significantly different from the existing work due to the existence of the visiting probability $\xi$. We believe our method introduces a novel method to deal with the total variance in the offline setting and can be helpful for the future work in the related fields.
>
> We then address the technical concerns you have raised.
>
> * For the proof of Theorem 2:
>     We first would like to kindly remark that our construction is based on a known horizon $H$, so we only need to reason about the regularity of the terms introduced in our construction for a fixed $H$. Notice that, $c_1 \in (0, H)$ and $c_2 \in (0, c_1)$, as we only need $p_1 = 1 - \frac{c_1}{H}\in [0, 1]$ (Line 610). So we allow $c_2$ scales with $H$. Assume $c_1 = \kappa H$ where $\kappa \in (0, 1)$, then $c_2 = \left(\kappa - \sqrt{\frac{\kappa(1-\kappa)}{n(s_2, a)}}\right)H$ and $p_2 = 1- \frac{c_2}{H} = 1 - \kappa + \sqrt{\frac{\kappa(1-\kappa)}{n(s_2, a)}}$ (Line 611). We only need $\frac{1-\kappa}{\kappa} > n(s_2, a)$, i.e. $\kappa > \frac{1}{n(s_2,a) + 1}$ to satisfy all of the regularity condition and make our construction valid. We also want to emphasize that by our construction, Assumption 1 and 2 naturally hold, and we finally replace $n$ with $KH$ to get Theorem 2 stated in the main text.
>
> * For the bound of $\mathbb{V}_1(i)$:
>     + Your examples do not violate the conclusion:
>
>         We would like to kindly remark that, time-homogeneity does not play any important role to guarantee $\mathbb{V}_1(i)\leq H$. Instead, Assumption 1 indicates $\mathbb{V}_1(i)\leq H$ without any further assumptions, and if the cases you mentioned satisfy Assumption 1, then $\mathbb{V}_1(i)\leq H$ will hold.
>
>         For the case that $r_H = 1$, $r_h = 0$, $\forall h < H$, we can simply conclude that  $\forall \pi, h, s$, $V_h^\pi (s) \leq 1$, as no matter which state and time-step we start, we can only collect at most $1$ total reward. With $V_h^\pi(s)\leq 1$, we further know that $\mathrm{Var}(V_h^\pi(s))$  is upper bounded by $1$, and notice that $\sum_h \hat{\xi}_h^\pi = 1$, we can conclude that $\mathbb{V}_1(i) \leq H$. It's not clear when there exists a set of state that can only be reached at last step violates our conclusion.
>     + How Assumption 1 guarantees the conclusion:
>
>         In fact, Assumption 1 implicitly indicates that $V_h^\pi(s) \in [0, 1]$, $\forall \pi$, and $\forall s$ that can be reached by some policy $\pi^\prime$. This is due to the fact that we assume Assumption 1 holds *almost surely*. If such condition does not hold, then in the first $h$ step we take policy $\pi^\prime$ that can reach $s$ with non-zero probability, and accumulate the reward from step $h+1$ to $H$ with policy $\pi$ that can be larger than $1$, which means there can be some trajectories that have total reward larger than $1$ with non-zero probability, that contradicts to Assumption 1. With the conclusion we mention above, and notice that any state we can reach in $\widehat{\mathcal{M}}$ can be reached in $\mathcal{M}$ (as $\widehat{\mathcal{M}}$ is constructed with the data from $\mathcal{M}$), we can see if $\widehat{\xi}_h^\pi(s, a) > 0$ and $P(s^\prime|s, a) > 0$, then $V_h^\pi(s)\in[0, 1]$, which can simply lead to the result that $\mathbb{V}_1(i) \leq H$. We will make this more clear in the next revision.

---

> > ### Comment · Reviewer_vD82 · 2021-08-17
> > **Clarification requested on novelty of the contributions**
> >
> > I have read the other reviews and the authors' responses so far.
> > Thank you for the comments re: originality/novelty of the work.
> >
> > Looking through http://proceedings.mlr.press/v134/zhang21b their Technique 2 on Pg9 uses Assumption 1 (their Assumption 1 is stated on Pg5, and is identical to the assumption in the paper) to bound the total variance term. I think it is a clever idea.
> >
> > In the off-policy setting, it may be useful to adapt this idea, so I am definitely on-board with the motivation of the paper.
> > Using assumption 2, we guarantee sufficient coverage in the off-policy case meaning that \epsilon^\pi_h(s,a) >0 for all s,a. Isn't the current paper executing exactly Technique 2 with the added factor of \hat{\epsilon}^\pi_h(s,a) in each of the terms? Does Delta_1(i) in Line 245 just have the additional \hat{\epsilon}^\pi_h(s,a) factor to account for off-policy data, and the rest of the derivation in [27] goes through? Please correct me if I've misunderstood, I'd love to understand better how the visiting probability \epsilon interacts with the recursive variance bounding argument of [27] (especially when it is guaranteed to be non-zero for all states, actions; I can see it breaking when some \epsilon^\pi_h(s,a) = 0).
> >
> > Even if my understanding is correct (and thereby limiting the scope of novelty/originality of the contributions), I believe that demonstrating that the total variance recursion argument can work in off-policy settings is valuable. Acknowledging this can situate the current paper in the broader literature better. And I believe the paper can be even more impactful if it works with weaker notions of data coverage for Assumption 2 (but notions like \epsilon = d^\pi/d^\mu(s,a) could pose a non-trivial challenge to bound using the recursion since \epsilon(s,a) could now be >=0 rather than in (0,1]).

---

> > > ### Author Response · Authors · 2021-08-17
> > > **Clarification on our techniques**
> > >
> > > We would like to thank the reviewer for going through over our responses. We make some clarifications on our techniques based on the reviewer's comments in the following:
> > >
> > > * *On the $\xi_h^\pi(s, a)$ term*: We would first want to clarify that it's not sufficient coverage assumption indicates $\xi^\pi_h(s, a) >0, \forall (s, a)$. And in fact it's not held for all $(s, a)$. $\xi^\pi_h(s, a)$ is needed due to the intrinsic randomness of the transition in MDP, as well as the potential randomness of the policy. Even we use some deterministic policy $\pi$ at state $s$, then the randomness in transition (i.e. $P(s^\prime|s, \pi(s))$) will lead to different visitation to the next state. Thus it's necessary to introduce the visiting probability term $\xi^\pi_h(s, a)$ in the offline-scenarios. In contrast, [1] considered the online setting, which the regret is defined on each episode's trajectory. And thus they don't need to have the $\xi^\pi_h(s, a)$ in their analysis.
> > > * *Is our paper executing exactly Technique 2 in [1]?*: It's not exactly. First, notice that their paper consider online setting and want to bound the cumulative regret, but our paper instead consider offline setting and want to bound the error of the policy, which is already quite different and can be hard to directly compare our techniques with theirs. Secondly, we want to remark the term $\sum_s \mu(s)V_1(s)$ in our Lemma 1 can only emerge if we carefully utilize the MDP structure that eventually makes our analysis on off-policy improvement work. The similar term does not appear in [1]. Also, the term $M_2$ their refer to the summation of the upper bound over all of the episodes, but our error here can be naturally embedded to the whole recursion analysis. We admit that the idea of recursively bounding the variance is quite similar to [1], but the technical details are quite different.
> > > * *When some $\xi^\pi_h(s, a)=0$*: In this case, our analysis still not breaks as we don't make any assumption on that, and our analysis still different from [1]. Still, they focus on the online setting, which is already quite different from the offline setting here.
> > >
> > > We hope our clarification can help answering some questions raised by the reviewer and we will revise our paper based on your suggestions.
> > >
> > > [1] Zihan Zhang, Xiangyang Ji, and Simon S Du. Is reinforcement learning more difficult than ban- dits? a near-optimal algorithm escaping the curse of horizon. arXiv preprint arXiv:2009.13503, 2020.

---

> > ### Comment · Reviewer_vD82 · 2021-08-17
> > **Authors' response adequately address technical concerns**
> >
> > Thank you for clarifying my technical questions. I mis-read Line 610 and wrongly concluded that 0 <= c_1 <= 1, when in fact 0 <= c_1 <= H, and indeed there is no problem with the required setting of c_2 to witness the lower bound.
> > The response re: V_1(i) <= H was also helpful and should be incorporated into the paper.
> >
> > I am increasing my score since my main questions re: motivation and correctness have been addressed.

---

> > > ### Author Response · Authors · 2021-08-17
> > > **Thanks for your kind review and helpful discussion.**
> > >
> > > We would like to thank for your kind review and helpful discussion. We will revise our paper according to your suggestion.

---

> ### Author Response · Authors · 2021-08-09
> **Thanks for your detailed and constructive review. We then address the concern on significance below. (2/2)**
>
> For the question raised in the significance part, we first want to kindly argue that, for a theoretical work, obtaining the tightest results and developing new technical tools are indeed helpful for the whole community, and we believe our work can be helpful for the future work in the related fields, even without sufficient discussions from the empirical side. For the specific questions mentioned in the review, we answer them below:
> * *If the plug-in approach work well in practice as theory predicted:* We believe for the cases that satisfy our assumptions, e.g. tabular MDPs that satisfies Assumption 1, and linear MDPs that satisfies Assumption 1 and 3, the error will not have polynomial dependency on the horizon. However, in practice, probably some of our assumptions can be violated (e.g. not in tabular case), which make the performance hard to predict. However, our results can be helpful in understanding this problem, that can potentially motivated new algorithms in the future.
> * *Can the assumptions reliably hold?* Assumption 1 can be held by properly normalizing the total reward. And the path finding problem, which only receives a reward of $1$ when reaching state then transiting to an absorbing state always transit to itself, naturally holds Assumption 1. Assumption 2 itself is a definition of $d_m$, and we can always find a $d_m$ that makes Assumption 2 holds. Our lower bound also shows the importance of such $d_m$ term. We want to kindly remark that, in most of the existing literature, an assumption on ``data coverage'' is always needed, although sometimes with slight difference (e.g. [2] and [3] define this term via the coverage on the optimal policy). What theoretical guarantees we can achieve beyond data coverage assumption is still an open problem and out of the range of this paper.
>
> * *What about the other method like off-policy policy gradient?* As we have discussed in Line 256-258, if the algorithm can find the best policy on the empirical MDP, then our analysis still applies and finally leads to a nearly horizon-free bound.
> * *Anchor data assumption:* The anchor data assumption recently emerged in several theoretical paper like [2] and [3]. We include it only for theoretical interest instead of some practical consideration.
>
> We also thanks for pointing out such minor issues and typos and we will revise our paper based on your comments. We hope our response properly address your concerns to re-evaluate our paper.
>
> [1] Yin, Ming, and Yu-Xiang Wang. "Asymptotically efficient off-policy evaluation for tabular reinforcement learning." International Conference on Artificial Intelligence and Statistics. PMLR, 2020.
>
> [2] Liu, Yao, Adith Swaminathan, Alekh Agarwal, and Emma Brunskill. Off-policy policy gradient with state distribution correction. arXiv preprint arXiv:1904.08473 (2019).
>
> [3] Yin, Ming, Yu Bai, and Yu-Xiang Wang. Near-optimal offline reinforcement learning via double variance reduction. arXiv preprint arXiv:2102.01748 (2021).
>
> [4] Sample-optimal parametric q-learning using linearly additive features. In International Conference on Machine Learning, pages 6995–7004. PMLR, 2019.
>
> [5] Qiwen Cui and Lin F Yang. Is plug-in solver sample-efficient for feature-based reinforcement learning? arXiv preprint arXiv:2010.05673, 2020.

---

> > ### Comment · Reviewer_vD82 · 2021-08-17
> > **Authors' response adequately address significance concerns**
> >
> > I have read the other reviews and the authors' responses so far.
> >
> > Thank you for answering my questions re: significance. They help clarify the motivation of this work, and address my concerns about situating these contributions in the off-policy literature.
> >
> > - Re: Assumption 2, I wondered that if the data collection policy was uniform random on all actions but one, then d_m will trivially have to be 0 and we do not get a meaningful guarantee. Even with a sufficiently exploratory policy, since Assumption 2 is a statement about the actual collected data, in practice this may imply an infeasibly large n (e.g. some states with vanishingly small probability under the data collection policy). The sharper notion I hoped for matches the "coverage of optimal policy" for policy optimization (as in [2,3]) and "coverage of policy being evaluated" for off-policy evaluation.

---

> > > ### Author Response · Authors · 2021-08-17
> > > **Discussion on the new concern**
> > >
> > > We would like to thank the reviewer for going through over our responses. We address the new concern below.
> > >
> > > * We are afraid we don't quite understand the issue raised by the reviewer, so we would like to discuss based on our understanding of the reviewer's question.
> > >     + *If the data collection policy was uniform random on all actions but one*: In this case, $d_m$ is $0$ and indeed we cannot guarantee to find the optimal policy on the MDP, as implied by our lower bound. This is definitely possible. To see this, we can consider even the simplest bandit case. If there's an arm that we never pull, how can we know if this arm will have a good performance or not. However, we can still have some guarantee, if we remove such action and consider the new MDP, in which the $d_m$ is not $0$, and we can identify a policy that is comparable to the optimal policy in this new MDP, which can be satisfactory. This is similar to the idea argued by Reviewer PYbq and ND3y.
> > >     + *Assumption 2 is about the actual collected data*: We would like to direct the reviewer to Line 208-213, that by $K = \tilde{\Omega}(1/d_m)$ (notice that we assume this in all of our theorems), even we make the assumption only on the behavior policy, it can still be translated to the assumption on actual collected data up to some multiplicative constant. We don't think it makes a lot of difference here.
> > >     + *Implies an infeasibly large $n$*: We would like to argue that, this is necessary if our target is finding the optimal policy on the original MDP, as suggested by our lower bound Theorem 2 and Theorem 4. But if we want to instead use different metrics, e.g. finding the optimal policy on some sub-MDP, or if we want to instead make different assumptions, like "coverage of optimal policy" as mentioned, then we can define a different $d_m$ that can reduce the samples we need and our analysis can still be applied (as we replied to Reviewer ND3y). In our paper we choose the most standard notion of the regret for simplicity, and we will revise our paper to discuss on this more clearly.
> > >
> > > We hope our clarification can help solving the issue on the definition of $d_m$ and how to interpret our result on $d_m$.

---

> > > > ### Comment · Reviewer_vD82 · 2021-08-17
> > > > **On d_m and Assumption 2**
> > > >
> > > > I see that Rev.ND3y and I went off-the-rails at the same spot in the paper, and I think it is because of Lines 208-212. The text before it does not imply that d_m > 0 (and indeed the examples that I constructed and the coverage assumption that Rev.ND3y pointed out have d_m = 0).
> > > > Please include the discussion about the proposed fix (i.e., consider a sub-MDP such that d_m > 0 in the sub-MDP) to address the confusion around d_m.

---

> > > > > ### Author Response · Authors · 2021-08-17
> > > > > **We will revise our paper accordingly.**
> > > > >
> > > > > Thanks for the discussion. We will revise our paper accordingly, especially on the issues related to $d_m$.

---

### Decision · Program_Chairs · 2021-09-27

**Decision:**

Accept (Poster)

**Comment:**

According to the reviews, which are all in favor of accepting the paper (although most of them only slightly), this is a solid contribution for which I recommend acceptance. The points raised in the reviews and discussed in the rebuttal (e.g. concerning motivation and some of the debated technical details) shall be taken into account when preparing the final version.